# How Can Proteomics Help to Elucidate the Pathophysiological Crosstalk in Muscular Dystrophy and Associated Multi-System Dysfunction?

**DOI:** 10.3390/proteomes12010004

**Published:** 2024-01-16

**Authors:** Paul Dowling, Capucine Trollet, Elisa Negroni, Dieter Swandulla, Kay Ohlendieck

**Affiliations:** 1Department of Biology, Maynooth University, National University of Ireland, W23 F2H6 Maynooth, Co. Kildare, Ireland; paul.dowling@mu.ie; 2Kathleen Lonsdale Institute for Human Health Research, Maynooth University, W23 F2H6 Maynooth, Co. Kildare, Ireland; 3Center for Research in Myology U974, Sorbonne Université, INSERM, Myology Institute, 75013 Paris, France; capucine.trollet@upmc.fr (C.T.); elisa.negroni@upmc.fr (E.N.); 4Institute of Physiology, Faculty of Medicine, University of Bonn, D53115 Bonn, Germany; swandulla@uni-bonn.de

**Keywords:** dystrophin, dystrophinopathy, integromics, mass spectrometry, multi-omics, muscle proteomics, myofiber, myology, neuromuscular disease, omics

## Abstract

This perspective article is concerned with the question of how proteomics, which is a core technique of systems biology that is deeply embedded in the multi-omics field of modern bioresearch, can help us better understand the molecular pathogenesis of complex diseases. As an illustrative example of a monogenetic disorder that primarily affects the neuromuscular system but is characterized by a plethora of multi-system pathophysiological alterations, the muscle-wasting disease Duchenne muscular dystrophy was examined. Recent achievements in the field of dystrophinopathy research are described with special reference to the proteome-wide complexity of neuromuscular changes and body-wide alterations/adaptations. Based on a description of the current applications of top-down versus bottom-up proteomic approaches and their technical challenges, future systems biological approaches are outlined. The envisaged holistic and integromic bioanalysis would encompass the integration of diverse omics-type studies including inter- and intra-proteomics as the core disciplines for systematic protein evaluations, with sophisticated biomolecular analyses, including physiology, molecular biology, biochemistry and histochemistry. Integrated proteomic findings promise to be instrumental in improving our detailed knowledge of pathogenic mechanisms and multi-system dysfunction, widening the available biomarker signature of dystrophinopathy for improved diagnostic/prognostic procedures, and advancing the identification of novel therapeutic targets to treat Duchenne muscular dystrophy.

## 1. Introduction

The extraordinary cellular complexity and molecular heterogeneity of biological systems are formidable bioanalytical challenges for the systematic survey of dynamic processes during physiological adaptations under healthy conditions versus pathophysiological alterations in a diseased state. The multi-cellular organization of the average human body is estimated to contain over 400 different cell types that form a network of over 36 trillion cells, which in turn display trillions of molecules per average cellular unit, including diverse species of nucleic acids, proteins, carbohydrates, lipids, minerals and metabolites [1,2,3,4]. A meta-analysis of protein abundance distribution suggests that the average eukaryotic cell contains approximately 43 million proteins [5], making the systematic cataloging and differential analysis of all protein species in health and disease a daunting task in modern proteomics [6,7,8]. At the level of the hierarchical biological organization of highly complex biomolecular systems, the number of protein-coding genes has been determined to be approximately 20,000 individual human genes [9,10,11], which probably generate over a million dynamic proteoforms [12,13,14] due to extensive genomic variations, alternative RNA splicing and extensive post-translational modifications of protein products [15,16,17]. Proteoforms can be defined as the expressed variants of the protein products that are encoded by a single gene, whereby the different molecular forms are generated by genetic variations such as alternative promoter usage, alternative splicing of RNA transcripts due to mechanisms such as exon skipping, and extensive post-translational modifications, including proteolysis, phosphorylation and glycosylation [12].

This perspective article reviews and discusses the means by which proteomics can be employed in an optimum way to identify and characterize the individual proteoforms that are present in the technically accessible skeletal muscle proteome [18,19,20]. The main focus is on the proteomic survey of the molecular and cellular pathogenesis of a multi-system neuromuscular disorder [21,22], i.e., X-linked Duchenne muscular dystrophy (DMD) [23,24]. The abnormal expression of the full-length Dp427-M isoform of the membrane cytoskeletal protein dystrophin [25] that is encoded by one of the largest genes in the human genome, the *DMD* gene [26], is the underlying cause of this devastating disorder [27]. Although this monogenetic disease can be classified as a primary muscle-wasting disorder with a main defect in the membrane cytoskeleton [28], the progressive decline of contractile strength in Duchenne patients is accompanied by body-wide alterations [29,30,31] and multi-system dysfunction [32,33,34]. This review briefly summarizes key results from systematic proteomic studies with special reference to muscular dystrophy research. A description of the routine usage of top-down proteomics versus bottom-up proteomics in basic and applied myology, including a discussion of bioanalytical advantages versus technical challenges, is provided. Besides studying the molecular pathogenesis of dystrophinopathy and being an irreplicable tool for biomarker discovery, proteomics is also highly suitable for the identification of novel therapeutic targets and the elucidation of drug mechanisms [34].

Based on the high-throughput and large-scale methodology that is currently available for proteomic applications, a future systems biological approach is outlined for the holistic characterization of dystrophinopathy. This would include the amalgamation of findings from multi-omics studies, using mass spectrometry (MS)-based proteomics as the core discipline for conducting protein biochemical surveys, to achieve a high degree of integromic data handling [35]. A key bioanalytical aspect would be to assess pathophysiological crosstalk between individual organ proteomes in muscular dystrophy that could include the evaluation of the role of muscle–bone interactions, brain–muscle signaling, the metabolic liver–fat–muscle axis, the influence of muscular alterations on the kidneys, gastrointestinal tract and the cardio-respiratory system, as well as the linkage of muscle changes to immune responses. Verification studies to evaluate the validity of multi-omics-based networks and their involvement in disease processes would be carried out by robust and standardized physiological, molecular biological, biochemical, immunochemical and histological assays. In the long term, integrated proteomics could form the scientific basis for developing a more complex understanding of neuromuscular pathogenesis and associated multi-system dysregulation, as well as be helpful in identifying novel biomarker candidates for the improved diagnosis, prognosis and therapeutic monitoring of muscle diseases and their body-wide complications.

## 2. Mass Spectrometry-Based Proteomics: Top-Down versus Middle-Up/Down versus Bottom-Up Approaches

Proteomics can be used for the biochemical identification of individual protein species, the detailed characterization of peptides and proteins and their post-translational modifications, and the systematic cataloging of entire proteomes, as well as for comparative studies of complex protein mixtures [36,37,38]. Mass spectrometric surveys of proteins were initiated in the late 1990s [39,40,41] and have been greatly refined over recent decades to the current state of single-cell resolution [42,43,44]. Current proteomic analysis pipelines, which generally consist of sample preparation, protein extraction, efficient protein separation, different degrees of controlled protein fragmentation, mass spectrometric analysis, bioinformatic assessment and independent verification studies, can be categorized into two main types of approaches, i.e., top-down proteomics [45] versus bottom-up proteomics [46], plus an additional third category in the form of middle-up/down proteomics [47]. Figure 1 provides an overview of the major steps that are involved in routine proteomic analyses of proteins and their specific proteoforms.

The most frequently employed methods in skeletal muscle proteomics have recently been reviewed, including top-down approaches, bottom-up techniques, comparative studies and membrane protein analyses [48,49,50]. The main procedures are briefly summarized in the below subsections, which outline the basic rationale of top-down versus bottom-up proteomics, protein separation, sample handling, the importance of optimized protein digestion, mass spectrometric techniques and data acquisition. Detailed descriptions of the key techniques employed in MS-based proteomics are beyond the scope of this perspective article that instead focuses on the actual application of proteomics for a more in-depth understanding of the pathobiochemical aspects of the multi-system pathology of dystrophinopathy. Excellent reviews are available that provide a comprehensive and critical examination of the main analysis pipelines and most frequently applied methodologies in the field of proteomics research [45,46,47,51,52,53,54].

### 2.1. Top-Down Proteomic Approaches

Although liquid chromatography (LC) is currently the most frequently used method for large-scale and high-throughput protein separation prior to MS-based analysis, gel electrophoresis (GE) represents an excellent technique for the efficient preparation of isolated and intact proteoforms [55,56,57,58]. Within the portfolio of GE methodology, two-dimensional gel electrophoresis (2D-GE) is a significant and well-established technical platform [15,59,60,61] that allows users to perform a comprehensive top-down proteomic analysis [62,63,64]. Two-dimensional gel electrophoresis that uses isoelectric focusing (IEF) in the first dimension and standard sodium dodecyl sulfate polyacrylamide gel electrophoresis (SDS-PAGE) in the second dimension [65] enables the separation of complex proteomes from a variety of samples based on the unique combination of the isoelectric point (p*I*) and molecular mass of individual protein species [66]. These key parameters are then used to generate a map of proteins, representing changes in protein abundance levels of distinct proteoforms [15], and in some examples, post-translational modifications (PTMs) [67,68,69]. Two-dimensional gel electrophoresis facilitates the separation of hundreds to thousands of proteins on one gel [63], with some research groups having established protocols that allow for over 4000 proteins to be precisely separated with high accuracy [61,70]. Importantly, 2D-GE has been optimized to separate the skeletal muscle proteome [59,60,71].

A number of specialized gel stains are available, including SYPRO Ruby, Deep Purple, silver stain and Coomassie Brilliant Blue (CBB) for protein labeling [72,73,74,75,76], as well as protein species-selective procedures for the detection of glycoproteins, such as Pro-Q Emerald gel stain [77,78], or phosphoproteins, such as Pro-Q Diamond gel stain [79,80]. SYPRO Ruby is a highly sensitive (~1 ng) fluorescent stain that can accurately quantitate protein expression levels. Silver stain and CBB [72] are robust staining methods facilitating the quick visualization of results [73], while Pro-Q Emerald and Pro-Q Diamond are specific to glycosylated and phosphorylated proteins, respectively, conferring a degree of specialization with this approach. Trypsin is the gold standard for protein digestion [81]; however, alternative proteases, such as endoproteinase Glu-C, endoproteinase Asp-N or chymotrypsin, can help increase amino acid sequence coverage by generating unique peptides that are complementary to tryptic peptides [82,83,84,85].

The development of the differential imaging gel electrophoresis technique for increased sensitivity and reproducibility using fluorescent dyes proved to be a significant addition to sample analysis using 2D-GE [86,87,88]. Following optimized sample preparation [89], fluorescence two-dimensional difference gel electrophoresis (2D-DIGE) is based on labeling of protein within a sample with a different fluorophore (CyDye3, CyDye5 or CyDye2) that binds covalently with the epsilon amino group of lysine residues for minimal labeling [90]. Typically, the internal control is a combination of all samples that will be analyzed within a single experiment and is labeled with CyDye2. The CyDye3- and CyDye5-labeled samples can then be normalized to CyDye2 for the identification of protein spots with different abundance levels when comparing samples [91,92,93]. Software packages including DeCyder, SameSpots and Dymension 3 can be incorporated into the workflow to aid in the identification of significant proteins [94,95]. Saturation labeling using CyDye3 and CyDye5 fluorophores with maleimide chemistry can be employed to label all cysteine residues within the sample of interest [96], enhancing sensitivity compared to the minimal approach [90]. As an alternative to gel-based protein separation, multi-dimensional protein identification technology (MudPIT) [97] can be employed, which is based on two-dimensional liquid chromatography (2D-LC) followed by MS-based analysis [98]. Extremely large proteins, such as the class of giant muscle proteins (e.g., titin, nebulin, obscurin, plectin, dystrophin and the ryanodine receptor Ca^2+^-release channel) [99], which do not properly enter the second dimension of conventional 2D gels due to their high molecular mass, can be separated by agarose 2D-GE [100,101,102] or one-dimensional gradient gel electrophoresis–liquid chromatography (GeLC) [103,104,105]. Recently, Melby et al. [106] described a highly sensitive top-down proteomic approach for the characterization of single myofiber heterogeneity using a one-pot protein extraction and sample processing strategy. The fraction with extracted muscle proteins was separated by a low-flow capillary LC method, which was coupled to a microflow multi-emitter nanoelectrospray source for optimum ionization efficiency prior to the MS-based analysis of intact proteoforms [106].

### 2.2. Bottom-Up Proteomic Approaches

Bottom-up proteomics refers to the characterization of proteins by analysis of peptides created from the protein through proteolysis [46,107]. An important initial consideration is whether fractionation is necessary or whether a more global analysis is the best approach. Many strategies are available to fractionate samples from relatively small quantities of cells/tissues in a short period of time, generating cytoplasmic, plasma membrane, nuclear, mitochondrial and cytoskeletal fractions as an example of subproteomics [108,109,110]. Column chromatography is also a successful approach, separating proteins based on size, charge or affinity. Isolation of single cells for MS-based analysis has the ability to identify the emergence of cellular heterogeneity and distinct cellular mechanisms underlying pathophysiological processes [111] and has been successfully applied to the study of single myofibers [112].

For bottom-up proteomic approaches, both in-solution and filter-based approaches are routinely used for sample preparation [113,114]. The filter-aided sample preparation (FASP) technique [115] and the suspension trapping (S-Trap) method [116] are popular for peptide generation. Enrichment strategies can be used to isolate and enrich sub-populations of peptides based on specific chemical properties [117,118]. A description of the analysis pipelines for investigating PTMs specific to skeletal muscle cell biology has recently been collated [48]. Phosphoproteomics has become one of the most active research areas in proteomic studies with phosphopeptide enrichment being a critical step in the analysis [119,120]. In addition to dedicated antibodies (pTyr antibody), phosphopeptides are enriched by their selective interaction with metals in the form of chelated metal ions or metal oxides (ferric nitrilotriacetate/Fe-NTA and TiO_2_ immobilized resins) [121]. Protein ubiquitination is a dynamic multifaceted PTM involved in many key processes of physiology and pathophysiology. The enrichment of ubiquitinated peptides containing ubiquitin remnants can be achieved by antibody-based approaches that specifically recognize the di-glycine motive remaining after digestion with trypsin [122]. In bottom-up proteomics, 1D to 3D LC methods are routinely used approaches in separating peptides prior to MS analysis [123].

### 2.3. Mass Spectrometric Analysis and Data Acquisition Techniques

Untargeted label-free quantitation (LFQ) of proteins aims to determine the relative abundance of peptides/proteins when comparing multiple biological samples [124,125,126]. LFQ has been successfully integrated into single-cell proteomic workflows [127]. For LFQ, two major approaches are routinely used, i.e., spectral counting and measuring MS1 signal intensities. In contrast to LFQ-based analyses, quantitative label-based techniques [128] are carried out with isobaric tagging for relative and absolute quantitation (iTRAQ) [129], stable isotope labeling by amino acids in cell culture (SILAC) [130], isobaric tandem mass tagging (TMT) [131], isotope-coded protein labeling (ICPL) [132] and isotope-coded affinity tags (ICAT) [133]. With SILAC experiments, proteins are metabolically labeled by culturing cells in media containing normal and heavy isotope amino acids, which are distinguishable by MS when samples are mixed together in equal ratios prior to analysis [130]. Alternatively, in vivo SILAC can be used to study whole organisms, such as mouse models of dystrophinopathy [134]. Multiplexed quantitative proteomics using the TMT technique is an unbiased quantification approach that can be adopted when evaluating peptide/protein abundance levels in a multitude of sample types [135].

Within the MS instrument, fragmentation approaches include the use of collision-induced dissociation (CID), higher-energy collisional dissociation (HCD) and/or electron transfer dissociation (ETD), depending on the specific application [136]. CID has become a routine approach for the fragmentation of peptides for protocols involving LFQ, while ETD is seen as the method of choice for peptides carrying labile PTMs [137].

A number of data acquisition techniques are available when using MS analyses, the two most widely used are data-independent acquisition (DIA) [138] and data-dependent acquisition (DDA) [139]. During MS/MS analysis, MS2 spectra are produced from the fragmentation of a product ion in a particular m/z range, following the operation of MS1 in scan mode. In DIA, effectively all peptides are fragmented together, resulting in complex MS2 spectra, but the values are comprehensive across the run time. Spectral libraries are employed to extract information from the wide-ranging data, facilitating quantification at the MS2 level. Sequential Window Acquisition of all Theoretical Mass Spectra (SWATH) [140] is a specific variant of the DIA approach [141], facilitating a deep proteome analysis, where all ionized peptides that exist within a specified mass range are fragmented in a systematic manner [142].

Table 1 lists major proteomic techniques, as well as a variety of biochemical, immunochemical and cell biological methods that can be used to independently verify the MS-based identification of specific proteoforms or the findings of changes in protein abundance from comparative proteomic studies. These methods are frequently employed to study cell or tissue specimens derived from normal versus diseased skeletal muscles, including the following:Two-dimensional gel electrophoresis (2D-GE) [55,56,57,58,59,60,61,62,63,64,65,66,71];Differential imaging gel electrophoresis (2D-DIGE) [86,87,88,89,90,91,92,93];Specialized gel-based methods for studying specific protein species [67,68,69,77,78,79,80];Gel electrophoresis–liquid chromatography methods (GeLC) [103,104,105];Protein microarrays [143,144,145,146];Sample preparation for proteomic analysis [89,113,114,115,116];Matrix-assisted laser desorption/ionization time-of-flight (MALDI-TOF) mass spectrometry [70,147,148];Surface-enhanced laser desorption/ionization time-of-flight (SELDI-TOF) mass spectrometry [149,150];Liquid chromatography–tandem mass spectrometry (LC-MS/MS) [54,151,152];Label-free quantification (LFQ) mass spectrometry [124,125,126,127];Isobaric tandem mass tagging (TMT) [128,131];Stable isotope labeling by amino acids in cell culture (SILAC) [130,134];Isobaric tagging for relative and absolute quantitation (iTRAQ) [128,129];Native mass spectrometry [153,154,155];Microproteomics using laser capture microdissection [156,157,158];Single-fiber proteomics [112,159];Enzyme-linked immunosorbent assay (ELISA) [160,161,162];Immunoblot analysis [163,164,165];Simoa bead-based immunoassay (SiMoA) [166,167,168];Microscopical analysis [169,170,171,172] including imaging mass cytometry (IMC) that utilizes metal-labeled antibodies [173,174,175];Flow cytometry [176,177,178];Protein interaction assays [179,180,181];Enzyme assays [182,183,184].
proteomes-12-00004-t001_Table 1Table 1Overview of key proteomic and biochemical techniques that are commonly used to study skeletal muscles in health and disease.ApproachBioanalytical AdvantagesDisadvantages/LimitationsReferencesTwo-dimensional gel electrophoresis (2D-GE)Two-dimensional gel electrophoresis has the ability to separate intact proteoforms. Preset conditions, such as pH ranges, size of the 2D gel and staining methods, can be adjusted to increase resolution. Straightforward to interface with many powerful biochemistry techniques including immunoblotting. Two-dimensional gels can be imaged using stains or fluorescent dyes, including labeling of PTMs.Two-dimensional gel electrophoresis exhibits a narrower dynamic range as compared to certain LC-based separation methods. Difficult to resolve very acidic or very basic proteins. Problematic analysis of very low- or extremely high-molecular-weight proteins.[55,56,57,58,59,60,61,62,63,64,65,66,71]Differential imaging gel electrophoresis using fluorescence two-dimensional difference gel electrophoresis (2D-DIGE)Two-dimensional difference gel electrophoresis allows the simultaneous investigation and comparison of three different samples on one two-dimensional gel, thus reducing gel-to-gel variability. Normalization within an experiment can be carried out via the inclusion of an internal control (such as CyDye2) in all sample sets.A significant number of steps are involved, taking multiple days to complete. Multiple phenotype comparison is still a challenge using the 2D-DIGE technique.[86,87,88,89,90,91,92,93]Gel electrophoresis–liquid chromatography (GeLC) methodsThe initial 1D-GE step using the GeLC-MS/MS technique allows for the efficient separation of extremely large proteins that do not properly separate in conventional 2D gels.GeLC-based methods are based on crowded 1D gel bands with the limited resolution of individual protein species.[103,104,105]Protein microarraysMicroarray technology allows high throughput of samples. Different formats are available for general and targeted custom screening approaches. Systems can be arrayed as semi-quantitative or quantitative formats.High-quality antibodies are not available for all targets. Microarrays require two specific antibodies for each target from the specific sample of interest.[143,144,145,146]Matrix-assisted laser desorption/ionization time-of-flight (MALDI-ToF) mass spectrometryMALDI-ToF MS is characterized by a simple operation mode and good mass accuracy, as well as high resolution and sensitivity for peptide mass fingerprinting (PMF). The method can be used for profiling and imaging of proteins directly using thin tissue sections (MALDI-IMS; imaging mass spectrometry)The sequence information provided by MALDI-ToF MS is generally not as comprehensive as that generated by LC-MS/MS. The method has reduced success rates for identifying proteins that are not in databases.[70,147,148]Surface-enhanced laser desorption/ionization time-of-flight (SELDI-ToF) mass spectrometrySELDI-ToF MS allows for high-throughput analyses. The preanalytical sample preparation is rapid and streamlined due to the ability to achieve chromatographic separation using a variety of protein-chip surfaces.Results are generally based on peptides and smaller proteins (<30 kDa). Additional effort is required to identify peaks of interest. Relatively low resolution of MS scans and low sensitivity.[149,150]Label-free quantification (LFQ) mass spectrometryLFQ MS analysis does not require expensive chemicals or metabolic tags, making it a cost-effective proteomic method. The time needed for sample preparation is significantly reduced due to a straightforward workflow as compared to labeling techniques.Factors such as the peptide or spectral count have limitations. Considerably more LC-MS time is needed for sample analysis. Low-abundance peptides may be underrepresented.[124,125,126,127]Isobaric tandem mass tagging (TMT)The TMT method makes it possible to analyze a significant number of samples that can be labeled (18-plex). Specifically linked protocols, such as synchronous precursor selection (SPS), can be helpful in identifying and quantifying low-abundance proteins.During TMT experiments, replication in labeling procedures and batch effects cannot be completely uniform. The method is associated with a high cost of reagents.[128,131]Stable isotope labeling by amino acids in cell culture (SILAC)For the SILAC approach, no in vitro labeling steps are necessary in the experimental procedures. Heavy and light amino acids share the same physico-chemical properties.SILAC has limited sample multiplexing capabilities and can only be carried out using cell culture or labeling of whole organisms.[130,134]Multiplex enzyme-linked immunosorbent assay (ELISA)The ELISA method can be conveniently used for the verification of the proteomic identification of distinct protein species. Multiplex ELISA techniques use fewer wells and/or plates for sample handling and have increased throughput capabilities and the ability to develop custom panels.Identifying antibodies with high specificity is a challenge due to issues with cross-reactivity. Proteins that are present at different abundance levels make linearity over a wide range of concentrations difficult.[160,161,162]Microscopical analysisHistological, histochemical and immunofluorescence microscopical studies can be employed to confirm proteomic results. Verification analyses can be carried out with both freshly dissected or frozen tissue samples for single-cell analysis. The techniques allow the subcellular localization of protein expression levels in a tissue sample with a fast turn-around time to achieve meaningful results. Of note, the recent development of imaging mass cytometry using metal-labeled antibodies has greatly improved the scope of microscopical investigations.Although these techniques provide data on the single-cell level, the optimization and quantifying results can be difficult. Immuno-histological studies can be subject to human error. Often, a highly trained histopathologist is needed for the proper interpretation of results. Imaging mass cytometry is associated with high costs due to the production of special antibodies.[169,170,171,172,173,174,175]Flow cytometry (FC)FC allows simultaneous cell biological analysis with multiple parameters. The method identifies small populations of cells within complex samples and allows for the quantification of fluorescence intensities.For a successful analysis, the method requires the careful choosing of a suitable combination of fluorochrome conjugates. Complex instruments are prone to analytical problems.[176,177,178]


## 3. Skeletal Muscle Heterogeneity and Muscle Proteomics

Skeletal muscle heterogeneity, cellular complexity and multi-systems crosstalk present serious bioanalytical challenges for systematic proteomic and biochemical studies in basic and applied myology. The skeletal muscle proteome can be defined as the totality of all protein species, i.e., proteoforms and their dynamic PTMs, that are produced at a given time by the muscle-specific expression profile of the genome [185]. It is crucial to keep in mind that skeletal muscle tissues are markedly heterogeneous in their cellular composition. This cell biological fact has to be taken into account during omics-type analyses, including comparative skeletal muscle tissue proteomics [18]. Proteoform heterogeneity due to genomic, post-transcriptional and post-translational effects has a profound influence on skeletal muscle proteome diversity [19,186].

### 3.1. Tissue Heterogeneity and Cellular Complexity of Skeletal Muscles

Multi-nucleated myofibers contribute to 47% of the biomass in the average human body [4] and are involved in diverse biological functions, such as voluntary movements, posture, bioenergetic and metabolic integration, muscle–skeletal balance, bodily protection, the regulation of thermogenesis, respiration, communication and the provision of an abundant protein reservoir during extended periods of starvation [187,188,189]. The proper physiological functioning of the voluntary contractile system is based on highly coordinated interactions between the central and peripheral nervous systems on the one hand and the various subtypes of skeletal muscles that are associated with a network of capillaries, elaborate layers of extracellular matrix (ECM) and embedded satellite cell populations on the other hand [188]. Individual muscles usually contain a mixture of slow and fast fiber populations plus hybrid fibers. The main biological features that can be used to differentiate between the main myofiber types are histological, physiological, biophysical, metabolic and biochemical properties [190].

Slow-oxidative type I fibers are characterized by small-size motor neurons, relatively small cell diameter, high capillary density, prominent mitochondrial density, slow contraction time with low levels of force production, aerobic activity with high oxidative capacity based on fatty acid oxidation, and low glycolytic metabolism, which is associated with high resistance to fatigue [191]. In contrast, faster-contracting type II fibers exhibit larger-size motor neurons, larger cell diameter, lower capillary density, lower mitochondrial density, fast contraction time with high levels of force production, predominantly anaerobic activity with low-to-intermediate oxidative capacity, and high glycolytic activity based on the metabolization of glucose/glycogen, which is associated with low resistance to fatigue [191]. Type II fibers can be further subdivided into fast-oxidative/glycolytic type IIa and fast-glycolytic type IIx myofibers in mature human muscles [192]. The category of hybrids ranges from type I/IIa to type IIa/IIx myofibers. In small rodents, which are frequently used as animal models in muscular dystrophy research [193], a third subtype of extremely fast-twitching myofibers is present in their musculature, which is classified as type IIb [191].

### 3.2. Protein Markers of Myofiber Specification

Reliable biochemical/proteomic markers for the different myofiber types are represented by the isoforms of the contractile protein myosin II. Myosin heavy chains (MyHC) of the types MyHC-1/beta (*MYH7* gene), MyHC-2a (*MYH2* gene), MyHC-2x (*MYH1* gene) and MyHC-2b (*MYH4* gene) are superb indicators of the main fiber types I, IIa, IIx and IIb, respectively [190,191,192]. Developing and specialized types of skeletal muscles are also associated with particular myosin isoforms, such as MyHC-embryonic (*MYH3* gene) and MyHC-neonatal (*MYH8* gene) during myogenesis [194,195] and MyHC-eom (*MYH13* gene) and MyHC-15 (*MYH15* gene) in extraocular muscles [196,197]. A detailed review has recently outlined myosin isoform diversity in the contractile apparatus of skeletal muscles [49]. Fiber type specification is highly plastic, and together with the overall regulation of muscle tissue mass, is heavily influenced by neuromuscular activity levels, load bearing, diverse hormonal effects and nutritional supply [198,199,200]. A variety of histological, histochemical, immunofluorescence microscopical and biochemical methods is routinely used for fiber typing [169,170,171,172,201] but has recently been superseded by more sophisticated and high-throughput proteomic methodology [106,112,202,203].

### 3.3. Cellular Complexity of the Muscle Environment

The cellular environment of contractile fibers is highly complex and contains myogenic stem cells (MuSCs), mesenchymal stromal cells (MSCs), such as fibro/adipogenic progenitors (FAPs), and resident macrophages [204,205,206]. Muscle-specific satellite cells are positioned between the sarcolemma membrane and basal lamina. Following physical injury or disease, the MuSC pool is activated [207]. High levels of cellular proliferation are involved in the self-renewal of the satellite cell population, and differentiation produces myogenic precursor cells for repair mechanisms [208]. Biochemical markers of inactive satellite cells, activated myogenic cells, myoblasts and fused myotubes are differential expression patterns of CD34 and the transcription factors Pax7, Foxo, MyoG/myogenin, MyoD1 and Myf5 [209,210,211]. 

The complex cellular arrangement of the mature neuromuscular system creates a protected, regeneratable and highly flexible pool of motor units per individual skeletal muscle. The extrafusal myofiber population that forms a distinct motor unit usually receives its innervating signals from the axonal branches of a single α-motoneuron. Thus, the crucial muscle–nerve connections regulate the patterns of excitation–contraction–relaxation cycles. Coordinated contractions depend on the joined forces that are generated by all motor units forming a physiological motor pool within a single skeletal muscle [212]. Therefore, crude extracts from muscle biopsies are heterogeneous in composition. Of note, due to the specific nature of muscle tissues, which are characterized by large and elongated myofibers, a high abundance of sarcomeric structures and several layers of ECM, in combination with associated issues of subcellular fractionation and protein extraction procedures, only a near-to-complete coverage of the skeletal muscle proteome is currently possible [213].

### 3.4. Multi-System Interactions of Skeletal Muscles and the Muscle Secretome

Crucial physiological parameters of contractile muscle tissues are multi-systemic interactions that are provided by body-wide skeletal muscle signaling involving myokines [214], in combination with a diverse array of cytokines, hormones, osteokines, adipokines and growth factors [215,216,217]. Myokines can be defined as peptides or proteins that are released or secreted by skeletal muscles into the circulatory system and exert autocrine, paracrine and endocrine effects. Hence, these muscle-derived signaling factors influence the muscle itself at the local level, as well as trigger changes in short- or long-term distant cells/tissues/organs [214]. The secretome of skeletal muscles has been characterized by MS-based proteomics [218] and a large number of myokines have been identified in developing, regenerating and matured muscles [219,220,221]. Major patterns of organ crosstalk and signaling axes are summarized in Figure 2, including the brain, skeletal muscle, heart, bone, liver, gut and fatty tissue [214,215,216,217].

An excellent example of how crosstalk between skeletal muscles and other organ systems can trigger severe pathophysiological changes is muscle-associated rhabdomyolysis. This disorder can be triggered by diverse initiators, such as crush injury, sepsis, drug overdose or extreme physical exertion [222], as well as being based on genetic abnormalities [223]. During an episode of rhabdomyolysis, the disintegration of skeletal muscles releases a large number of proteins and electrolytes into the circulatory system, causing downstream dysfunction that may lead to kidney failure and heart fibrillation. Renal abnormalities are often linked to aggregates of large amounts of released muscle myoglobin and cardiac issues are associated with elevated levels of K^+^ ions in the circulatory system [224]. The abnormal brownish staining of urine due to the massive release of muscle proteins is a key diagnostic indicator of rhabdomyolysis. Therefore, the proteomic profiling of urine suggests itself as an ideal non-invasive diagnostic tool to monitor the extent of body-wide effects due to the release of muscle-associated proteins during an acute episode of rhabdomyolysis [225], which can be life-threatening [222].

### 3.5. Skeletal Muscle Proteomics

The proteomic profiling of skeletal muscles is concerned with both the detailed intra-proteomic analysis of the heterogenous cell types that are present within the neuromuscular system and inter-proteomic surveys focusing on the crosstalk between skeletal muscles and other organs via the circulatory system. Based on the dynamic nature of proteoforms as the basic units of the proteome [12,13,14,15], skeletal muscle proteomics attempts to increase our biochemical and pathophysiological knowledge of protein changes in health and disease [18,19,20,48]. However, since skeletal muscle tissues are heterogeneous in their cellular composition, this biological fact is reflected by the diverse constitution of the accessible muscle proteome [185]. Skeletal muscles contain both tissues that are difficult to homogenize and comprise a large amount of sarcomeric proteins, membrane-associated proteins and relatively insoluble proteins of the ECM, making the proteomic analysis of total extracts a difficult task. Proteome-wide effects on skeletal muscles are routinely studied using cultured muscle cells, human biopsy specimens and muscles derived from animal models [48]. The collection of mass spectrometrically identified and muscle-associated proteins consists of over 10,000 individual species. The members of the core muscle proteome have been established by the systematic survey of various skeletal muscle types [226,227,228,229,230,231,232] and focused analyses of slow versus fast subtypes of myofiber populations [233,234,235,236,237,238,239,240,241,242]. The MS-based analysis of skeletal muscle specimens is routinely carried out with crude total extracts or subcellular fractions using sophisticated biochemical separation methodology. Figure 3 summarizes the subdisciplines involved in the integrative proteomic analysis of the skeletal muscle system.

It is important to stress that changes in non-muscle phenotypes in muscular dystrophies are most likely due to a combination of both organ crosstalk via the circulatory system and intrinsic changes within individual tissues/organs. This is especially relevant to DMD, as it is known that the *DMD* gene contains several promoters that produce eight different tissue-specific dystrophin isoforms, as outlined in the below section on the genetic basis of dystrophinopathy. The tissue-specific expression of dystrophins can be differentially affected by various mutations in the *DMD* gene. Thus, not all non-muscle effects are due to organ crosstalk but can be based on mutation-specific alterations in non-muscle tissues.

The isolation of distinct muscle fractions most frequently involves the microdissection of cellular structures or optimized tissue homogenization followed by differential centrifugation, density gradient ultracentrifugation and/or biochemical affinity isolation approaches [48,49,50]. Affinity purification can be carried out with pharmacological agents, immobilized lectins or suitable antibodies. The most intensively studied subcellular fractions derived from myofibers are the sarcolemma, transverse tubules, sarcoplasmic reticulum (longitudinal tubules and terminal cisternae region), triad junctions, mitochondria (outer membrane, contact sites, inner membrane and matrix), nuclei, ribosomes, Golgi apparatus, lysosomes, peroxisomes, proteasome, sarcosol, the sarcomeric acto-myosin apparatus, the auxiliary titin and nebulin filament structures, the intracellular cytoskeletal network, costameres, and the various layers of the ECM, including the basal lamina, endomysium, perimysium and epimysium [213]. Single-myofiber proteomics, which presents a specialized form of single-cell proteomics [44] that is applied to the MS-based analysis of contractile fibers, is becoming increasingly important in the field of basic and applied myology [112,202,203,237,238,239].

## 4. The Pathoproteomic Profiling of Duchenne Muscular Dystrophy

### 4.1. The Genetic Basis of Dystrophinopathy

Primary abnormalities in the *DMD* gene are the underlying cause of dystrophinopathies [28], a group of progressive muscle-wasting diseases that include the severe Duchenne type of muscular dystrophy in early childhood [23,24] and the more benign and later-onset Becker’s muscular dystrophy [243]. This classifies dystrophinopathies as monogenetic diseases that are characterized primarily by chronic muscle wasting due to a main cellular defect in the membrane cytoskeleton. The degeneration of muscle fibers affects almost exclusively males due to the location of the defective gene within the Xp21.2 region on the short arm of the X-chromosome [26]. Diverse types of genetic abnormalities were shown to be associated with dystrophinopathies, including splice site mutations, nonsense point mutations, missense point mutations and mid-intronic mutations, as well as small and large insertions, small and large deletions and large duplications [244,245,246].

The almost complete loss of the full-length dystrophin isoform Dp427-M is the initial trigger that causes progressive myofiber degeneration. However, the 79-exon-spanning *DMD* gene with its 2.4 million bases has a highly complex genomic organization consisting of several promoter regions that are involved in the production of eight different and tissue-specific dystrophins [23]. Dystrophin proteins are represented by the full-length versions Dp427-M (muscle), Dp427-B (brain) and Dp427-P (Purkinje cells) and the shorter isoforms Dp260-R (retina), Dp140-B/K (brain/kidney), Dp116-S (Schwann cells), Dp71-G (ubiquitous) and Dp45 (nervous system) [247]. This diversity of tissue-specific dystrophin isoforms, which are affected differentially by particular mutations in the *DMD* gene, and secondary effects of muscle disintegration on other organs cause body-wide alterations in muscular dystrophy [29,30,31] and multi-system dysfunction [32,33,34], as discussed in detail in the below sections.

### 4.2. Pathoproteomics of Chronic Muscle Wasting in Dystrophinopathy

In skeletal muscle, the almost complete deficiency in the full-length dystrophin protein triggers an unstable membrane cytoskeleton and the collapse of the dystrophin-associated glycoprotein (DGC) complex, consisting of dystroglycans, sarcoglycans, sarcospan, dystrobrevins and syntrophins [248,249,250,251]. The disturbed trans-sarcolemmal linkage, which is usually provided by tight dystrophin/dystroglycan interactions within the DGC [252,253,254] causes the loss of the organizing dystrophin node [254]. In healthy muscles, the dystrophin-containing node of the sarcolemma is defined as the integrating structure of the intracellular cytoskeleton, the central provider of lateral force transmission, the plasmalemmal hub of fiber stabilization and a key point for cellular signaling mechanisms at the myofiber periphery. In muscular dystrophy, the collapse of the dystrophin node results in impaired sarcolemmal integrity, reduced lateral force transmission at weakened costamere structures, and pathophysiological Ca^2+^ influx into the sarcosol, which in turn triggers an increase in proteolytic degradation of muscle proteins [255,256,257]. The main feature of the cellular pathogenesis of DMD is progressive myonecrosis, which is accompanied by chronic inflammation, reactive myofibrosis and an impaired regenerative capacity due to satellite cell dysfunction, as recently reviewed [258]. Skeletal muscles are characterized by the presence of extremely large proteins, i.e., giant proteins such as titin and nebulin [99], as well as many distinctly hydrophobic membrane proteins [50] and complex layers of mostly insoluble ECM components [259]. This makes proteomic studies of total tissue extracts technically challenging. The extensive listings of proteome-wide changes in dystrophic muscles have previously been published in extensive reviews. These articles have summarized the findings from systematic proteomic surveys using both patient biopsy material and various animal models of dystrophinopathy [260,261,262,263,264,265].

Individual studies have confirmed complex proteomic changes due to myonecrosis and myofibrosis, including altered expression levels of proteins involved in the organization of the cytoskeleton, maintenance of the ECM, energy metabolism, the cellular stress response and the excitation–contraction–relaxation cycle [260,263]. Bioanalytical approaches have employed both top-down proteomics and bottom-up proteomics with crude extracts and select subcellular fractions. The main methods used for the screening of dystrophic muscle specimens included 2D-GE, 2D-DIGE, immuno-precipitation, affinity purification, chemical cross-linking, liquid chromatography, ICAT, SILAC, iTRAQ, MudPIT, MALDI-ToF MS and LC-MS/MS analysis [172,264,265]. Please see the above section on proteomic technology for details on these methods and their analytical advantages versus technical limitations. Proteins with a significant change in abundance were identified in a variety of skeletal muscle types [264]. The most prominent and reproducibly identified proteins include the adenylate kinase isoform AK1, annexins, small heat shock proteins, desmin, vimentin, tubulins, collagens, calsequestrin, B-type lamin, myoferlin, dysferlin, ferritin, carbonic anhydrase isoform CA3, the fatty acid-binding protein FABP3 and various contractile proteins [134,148,180,232,266,267,268,269,270,271,272,273,274,275,276,277,278,279,280,281,282,283,284,285,286,287,288,289,290,291,292]. Independent verification analyses using comparative immunoblotting, enzyme assays, histochemistry and immunofluorescence microscopy were employed to confirm proteome-wide changes in dystrophic skeletal muscles [172].

### 4.3. Pathoproteomics of Multi-System Dysfunction in Dystrophinopathy

Besides chronic skeletal muscle wasting [23,24] and impaired neuromuscular transmission [293], Duchenne patients suffer from multi-system dysfunction [29,30,31] involving a variety of tissue and organ systems [32,33,34]. These body-wide abnormalities include cardiomyopathy [294,295,296], respiratory failure [297,298,299,300], liver atrophy [301,302], renal failure [303,304,305,306], bladder dysfunction [307,308,309,310] and gastrointestinal complications [311,312,313], as well as bone fragility [314] and scoliosis [315,316,317]. A subset of Duchenne patients suffers from neurological deficiencies that manifest themselves as neurodevelopmental delays, emotional disturbances, mental retardation and behavioral problems [318,319,320,321,322]. The main non-skeletal muscle organ systems that are affected in dystrophinopathy include the following:Central nervous system: cognitive impairments, attention deficit, altered emotions, impaired language, memory deficiencies and altered coordination;Peripheral nervous system: abnormal transmission at nerve–muscle connections;Cardio-respiratory system: late-onset cardiomyopathy, cardio-respiratory syndrome, respiratory insufficiency;Liver: enlargement, steatosis, fibrosis, atrophy and ectopic fat deposition;Renal system: kidney failure, cardio-renal syndrome, hyperfiltration, hypertension and ectopic fat deposition;Bladder: dysfunction of the urinary tract and bladder;Bone: increased risk of bone fragility;Spine: high risk of development of scoliosis;Gastrointestinal system: delayed gastric emptying and pancreatic dysregulation;Immune system: hyperactivity causing chronic inflammation, spleen adaptations.

Although the detailed proteomic analysis of multi-system dysfunction in dystrophinopathy clearly lags behind the exhaustive MS-based analysis of dystrophic skeletal muscles [260,261,262,263,264,265], a small number of studies have been initiated to establish a more comprehensive picture of whole-body effects due to dystrophin deficiency. The pathophysiological status of non-skeletal muscle tissues has been mostly studied with the help of established animal models of dystrophinopathy and has included the investigation of the heart, liver, kidney, stomach/pancreas interface, spleen and brain.

Proteomic surveys of dystrophin-deficient hearts have been carried out by both top-down/gel-based proteomics and bottom-up proteomics [323,324,325]. Cardioproteomics is an established field within the systems biological multi-omics approach to determine the underlying mechanisms of heart disease [326,327,328,329]. Focusing on X-linked muscular dystrophy, biochemical, physiological and proteomic studies have revealed substantial changes in key components of the contractile apparatus, ion-regulatory elements, proteins involved in the maintenance of the cytoskeleton, components that are central to the stabilization of the basal lamina, molecular chaperones that mediate the cellular stress response, and proteins that are associated with oxidative and glycolytic energy metabolism in cardiomyocytes [95,330,331,332,333,334,335,336,337,338]. These alterations in the cardiac protein constellation agree with the cardiomyopathic complications seen in dystrophic patients [294,295,296]. In contrast to skeletal muscles, the DGC is not restricted to the sarcolemma in the heart but also localizes to the transverse tubular membrane system in cardiomyocytes [339]. These cell biological differences are reflected on the biochemical level by a slightly different composition of the dystrophin complexome in voluntary myofibers versus myocardial contractile cells [333,340]. This might explain why laminin is not majorly affected in dystrophic skeletal muscles but was found to be reduced in the basal lamina of dystrophic heart cells [335]. Protein perturbations in the dystrophin-lacking heart can be linked to muscle cell degradation, interstitial fibrosis and inflammation [323]. Importantly, the dystrophinopathy-related dysregulation of the heart and adaptations in the cardiovascular system are probably linked to detrimental changes in the overall circulatory system [30].

A late-onset pathophysiological effect of poor circulation due to a chronically weakened heart could trigger a lack of sufficient oxygen and nutritional supply to organs such as the liver and kidneys [32,33,34]. The proteomic screening of the liver from the *mdx-4cv* model of dystrophinopathy revealed changes in proteins involved in the metabolism of carbohydrates, fatty acids and amino acids, as well as components that are associated with the cellular stress response, the regulation of ion homeostasis and biotransformation [341]. This agrees with the observed liver abnormalities in Duchenne patients [301,302]. A striking finding was the MS-based demonstration of elevated levels of FABP5, a major member of the large family of fatty acid-binding proteins [342]. Increased expression of the FAPB5 isoform was confirmed by immunoblot analysis and confocal microscopy. Of note, Sudan Black staining labeled fatty deposits in the liver of dystrophic mice. These cellular changes in *mdx-4cv* hepatocytes agree with high levels of FABP5 and suggest the occurrence of altered patterns of fatty acid transportation and ectopic fat deposition in the liver in DMD [341]. In analogy to hepatic tissue, the proteomic analysis of serum from the same dystrophic mouse mutant was characterized by a high concentration of FABP5 [343,344], making it a potential serum biomarker candidate for the monitoring of hepatic alterations in association with dystrophinopathy [264,265,345].

In addition, the renal system can be severely impaired in some patients suffering from DMD and this might also be linked to abnormal circulation [303,304,305,306]. The proteomic analysis of *mdx-4cv* kidneys showed similar findings as already described for the liver in dystrophic mice, i.e., elevated levels of distinct fatty acid-binding proteins [346,347]. In the case of the kidneys, the FABP1 isoform was identified by MS-based analysis in association with ectopic fat deposits [346]. These results on the liver FABP5 and kidney FABP1 isoforms do not directly demonstrate a pathophysiological link between the increase in FABP levels and the accumulation of intracellular fat deposits. However, they agree with the disturbed fat metabolism in DMD. The FABP3 isoform is present at high levels in the heart and skeletal muscles, and proteomics has identified it as a robust biomarker of aerobic capacity in muscle [231]. Dystrophic skeletal and cardiac muscle tissues exhibit a drastically reduced concentration of FABP3 [232,274,275,276,277,278,279,290,335], which is mirrored by elevated FABP3 levels in the serum of Duchenne patients and *mdx*-type mouse models [348,349,350,351]. Hence, the analysis of serum FABP3, FAPB5 and FABP1 isoforms can be useful for evaluating the shedding or release of excess fatty acid-binding proteins from muscle tissues, the liver and kidneys, respectively, in the dystrophic phenotype [342]. Proteomic biomarker discovery for multi-system changes in dystrophinopathy is discussed in more detail in the below section.

An extensive immune response is observed in X-linked muscular dystrophy, causing chronic inflammation in the skeletal musculature [352,353,354] which presents an excellent opportunity for therapeutic interventions [355,356,357,358]. Both, the innate immune system and adaptive immune responses intermingle in a complex relationship in dystrophinopathy [359]. Besides activation of resident macrophages within the skeletal muscular system, cellular invasion by immune cells includes the movement of regulatory T cells, CD4+ T cells, CD8+ T cells, natural killer cells, eosinophils and monocytes into degenerating myofibers as a response to muscular dystrophy [360,361,362,363]. In muscular dystrophy, splenic abnormalities include morphological alterations to lymph nodes in the white pulp region of the spleen and adapted pools of splenic inflammatory monocytes, which are associated with drastically elevated levels of immune cell migration from the splenic reservoir to damaged muscles [364,365]. Since the spleen represents a key secondary lymphoid organ whose biological functions are involved in antigen detection, antibody production and the efficient removal of abnormal erythrocytes [366,367], splenic abnormalities may have body-wide consequences. The linkage between dystrophic skeletal muscles and the lymphoid system was assessed by proteomics [124,368]. Significant changes in the *mdx-4cv* spleen were identified in proteins that participate in cellular signaling, metabolic pathways and cytoarchitecture [368]. The spleen is probably involved in a pathophysiological crosstalk with dystrophin-deficient myofibers [34].

Gastrointestinal dysfunction in muscular dystrophy [29,311,312,313] was evaluated by an MS-based study focusing on proteome-wide changes at the interface between the pancreas and the muscular stomach wall of the *mdx-4cv* model of DMD. Lower levels of dystrophin and its associated glycoproteins, as well as laminin, filamin and titin, suggested a loss of cytoskeletal integrity leading to abnormal smooth muscle function in the gut [369]. The below section lists promising biomarker candidates of the above-discussed non-skeletal muscle tissues in X-linked muscular dystrophy.

Psychosocial and psychological care is an integral part of the management of Duchenne patients [31]. Since a sub-group of dystrophic children is afflicted by complex neurodevelopmental and neurological deficiencies, it was of interest to study potential changes in the dystrophin-deficient brain proteome. The mouse brain is frequently used in biochemical, proteomic and cell biological studies in order to characterize the potential involvement of the central nervous system in neuromuscular diseases [370]. Proteomic screening of the *mdx-4cv* brain revealed a decreased abundance of syntaxin-1B and the syntaxin-binding protein STXBP1, which are involved in synaptic vesicle docking mechanisms at the pre-synaptic membrane and the regulation of neurotransmitter release from neurons [371,372]. Key brain proteins with an increased abundance were identified as the glial fibrillary acidic protein (GFAP), the annexin isoform ANXA5, the neuron-specific enzyme ubiquitin carboxyl-terminal hydrolase isozyme L1 and the neuronal cytomatrix protein bassoon. GFAP is a member of the intermediate filament system and is tightly associated with astrocytes in the central nervous system [373,374,375]. Thus, increased levels of GFAP in the *mdx-4cv* brain, which were demonstrated by proteomics, comparative immunoblotting and immunofluorescence microscopy [371], strongly suggest the presence of astrogliosis being part of the neurodegenerative process in DMD [34]. However, astrogliosis appears to be a frequent occurrence in the brain in response to general tissue damage [373], as was recently shown by the proteomic analysis of the *wobbler* mouse model of amyotrophic lateral sclerosis [376]. This reduces the diagnostic usefulness of GFAP as a specific biomarker of muscular dystrophy-related changes in the central nervous system but does not limit its general suitability as an astrogliosis marker protein. Proteomics suggests nevertheless that neuronal disturbances and reactive astrogliosis might play a central role in the molecular pathogenesis of brain abnormalities in dystrophinopathy [371] that may lead to mental retardation, behavioral problems, cognitive impairments, emotional disturbances, attention deficit, impaired language, memory deficiencies and altered coordination in DMD [377,378,379].

Figure 4 gives an overview of the complexity of the multi-system pathology of dystrophinopathy [34] and how MS-based analyses can be used for interproteomic profiling. A crucial aspect of future muscular dystrophy research should be the establishment of a more comprehensive picture of pathophysiological inter-organ crosstalk between the various bodily systems in dystrophic patients. Combing findings from organ proteomics with biofluid proteomics could help us better understand the involvement of (i) skeletal muscle–bone interactions, (ii) the brain–muscle signaling axis, (iii) metabolic integration at the level of liver, fat and muscle, (iv) muscle disintegration in the context of renal dysfunction and gastrointestinal problems, (v) the cardio-respiratory system, and (vi) the immune system. The prediction of the severity of the disease phenotype in individual Duchenne patients and potential effects on the whole body based on specific mutations would be an important future tool for clinical work [380]. The relevance of proteomic analyses is summarized in Figure 4. Interproteomic profiling should lead to a better comprehension of the pathophysiological complexity of muscle-associated changes in combination with multi-system effects for biomarker discovery, as further discussed in the below section.

### 4.4. Proteomic Biomarkers of Muscular and Multi-System Changes in Dystrophinopathy

Besides testing novel pharmacological strategies [257,381,382,383,384] and immunomodulatory interventions [355,356,357,358,364], a variety of innovative therapeutic approaches are currently evaluated in the field of muscular dystrophy [385,386,387,388,389], including exon skipping [148,390,391,392,393], genomic editing [394,395,396], codon read-through [397], gene replacement with the help of adeno-associated viral vectors [398,399,400,401,402], dystrophin substitution with its autosomal homolog utrophin [403,404,405], and muscle stem cell therapy [406,407,408,409]. Hence, for the optimum pre-clinical testing of new therapies, the clinical evaluation of diverse patient cohorts during the various phases of clinal studies/trials, and long-term therapeutic monitoring, robust and specific biomarkers for the routine screening of the status of dystrophic patients are required [144,264,265,410,411]. Ideal protein biomarkers would be measurable in a non-invasive, or at least minimally invasive, way and be suitable for repeated sampling procedures [325,345,410,411,412].

Importantly, biomarker detection should not be overly influenced by age, gender, ethnicity, circadian rhythm, seasonal impact, co-morbidities and supportive treatments. Of note, the determination of biomarkers should not be susceptible to the generation of high levels of false positives and false negatives. The assay system should be optimized for a proper balance between specificity and sensitivity to measure the biomolecule of interest. The clinical application of a biomarker signature, rather than the usage of a single and often not completely reliable marker molecule, can be advantageous to cover more than one particular aspect of a complex pathophysiological process that may considerably change over time [413]. Thus, biomarkers that are highly suitable for initial screening and differential diagnostic purposes might not be ideal for prognosis, the testing of potentially adverse side effects and extended periods of therapeutic monitoring.

The main subtypes of proteomic biomarkers include the following:Susceptibility markers: for risk assessment of potential disease initiation;Diagnostic markers: for initial detection of a specific disease process;Prognostic markers: for envisaging disease progression and adverse clinical events;Predictive markers: for differential patient screening and individual sensitivities;Pharmacodynamic markers: for assessing the body’s response to a specific treatment;Therapeutic monitoring markers: for the repeated assessment of disease status following therapeutic intervention;Safety-related biomarkers: for determining potential adverse side effects.

Of special interest are biofluid markers that can be easily accessed in a simple, cost-effective, safe, non-invasive and pain-free way, such as biomolecules that are present in sufficient abundance in urine or saliva, or at least be testable in a minimally invasive procedure using serum/plasma-associated biomarkers. The routine usage of these types of disease markers would have several advantages over invasive muscle biopsy procedures. Although the histological and histochemical analysis of muscle tissue biopsies is highly useful for determining the various cell biological aspects of pathological changes, the potential occurrence of fiber type shifting, reactive mechanisms such as fibrosis and alterations in cellular components of interest [169,170,171,172], needle or open biopsy procedures are often associated with higher costs, more complex harvesting and handling of tissue specimens, patient anxiety, tissue damage triggering inflammatory responses, potential infection and usually a lack of capacity for repeated sampling approaches [414,415,416,417].

Routinely used general muscle damage markers, assayed alone or in combination, include the muscle-specific isoform of creatine kinase, the carbonic anhydrase isoform CA3, troponin subunit TnI, myosin light chain MLC1, fatty acid-binding protein FABP3, myoglobin, aspartate transaminase, enolase, aldolase, lactate dehydrogenase, alanine aminotransaminase and hydroxybutyrate dehydrogenase [413,418,419,420]. An altered concentration of these types of proteins is often observed after crush injury or strenuous physical exercise but also in association with a variety of neuromuscular diseases, autoimmune disorders, toxic insults, body-wide inflammation, infectious diseases, sepsis, alcoholism and drug abuse. It is therefore imperative to identify more specific biomarkers of dystrophinopathy that are not changed by other types of physiological or pathobiochemical challenges to the neuromuscular system.

Figure 5 summarizes the findings from major biochemical and proteomic studies aimed at the identification of biomarker candidates that are characteristic of both tissue-related changes and the extent of the release of muscle proteins from dystrophic myofibers and their surrounding tissues and extracellular environment. The full names of abbreviated protein species are listed in the figure legend. The upper part of the diagram lists skeletal muscle tissue markers related to (i) the disintegration of sarcolemmal integrity and initiation of myonecrosis (members of the dystrophin complex, such as dystrophin, dystroglycans and sarcoglycans), (ii) abnormal Ca^2+^ homeostasis triggered by the influx of ions into the sarcosol through the damaged sarcolemma membrane and impaired luminal Ca^2+^ buffering (Ca^2+^-binding proteins calsequestrin and sarcalumenin), (iii) cycles of tissue regeneration and muscle repair (myoferlin, dysferlin, annexins, CD34 and cadherin-13), (iv) intracellular compensation of dystrophin deficiency by up-regulation of other types of cytoskeletal components (vimentin, desmin, and tubulin chains alpha-1B, alpha-1C, alpha-8 and beta-2A), (v) macrophage invasion and the triggering of chronic inflammation (cathepsin B, secreted phosphoprotein SPP1 and lysozyme), (vi) an increase in the cellular stress response by elevation of the abundance of small heat shock proteins (αB-crystallin/HspB5 and cardiovascular cvHsp/HspB7) and (vii) extensive reactive myofibrosis (matricellular protein periostin, collagens such as COL6, fibronectin, dermatopontin, biglycan, lumican and asporin).

Biochemical and proteomic studies have identified a large panel of potential muscle-derived biofluid markers, both as intact proteins or peptide fragments [421,422,423,424,425,426]. As listed in the lower panel of Figure 5, the myomesin isoform MYOM3, a marker component of the M-line in sarcomeres, was identified in the form of peptide fragments in serum samples [427] and various titin fragments were clearly detected in urine from both Duchenne patients and animal models of dystrophinpathy [428,429,430,431,432]. The half-sarcomere spanning protein titin of 3.8 MDa is the largest known protein in skeletal muscles [99] and its detection in urine indicates a massive disintegration of the auxiliary filaments of the sarcomere structure in dystrophinopathy [431]. The list of biofluid markers ranges from metabolic and modulating enzymes (adenylate kinase, creatine kinase, carbonic anhydrase, matrix metallo-proteinases, aspartate transaminase, alanine aminotransaminase, lactate dehydrogenase, malate dehydrogenase), metabolite transporters (fatty acid-binding proteins, myoglobin), sarcomeric proteins (troponins, myomesin, myosin light chains, titin fragments), signaling molecules (interleukins) to ECM proteins (fibronectin) (Figure 5). The passive shedding of large numbers of skeletal muscle proteins into the circulatory systems is indicative of the extent of sarcolemmal disintegration due to the collapse of the DGC. The overall changes in these biomarkers demonstrate the complexity of the cellular pathogenesis that is triggered by dystrophin deficiency and causes myofiber degeneration and loss of sarcolemmal integrity, followed by chronic inflammation and massive fibrosis of dystrophic muscles [422]. In our opinion, the most suitable minimally invasive and biofluid-associated biomarkers of DMD are represented by amino-terminal and carboxy-terminal TTN fragments of titin in urine, kallikrein KLK1 in saliva and the combination of CA3/FABP3/MYOM3/MDH2 in plasma/serum samples.

Multi-system changes in DMD are reflected by both proteomic changes in particular organs other than skeletal muscles and in biofluid markers that are characteristic of alterations in the heart, liver and kidneys. As already outlined above, striking proteomic changes in the brain included an increase in GFAP, an established marker of astrogliosis, as well as annexins and vimentin [371]. In contrast to dystrophic skeletal muscles that exhibited no major changes in laminin, this major component of the basal lamina was shown to be drastically reduced in a dystrophin-deficient heart [335]. The proteomic screening of the liver and kidney showed ectopic fat deposition in conjunction with elevated levels of the fatty acid-binding proteins FABP5 [341,342] and FABP1 [342,346], respectively. The analysis of the dystrophin-lacking stomach/pancreas interface revealed changes in the ECM protein repetin [369]. The proteome of the spleen was found to show increased levels of MMP9 and TGM2 in the dystrophic phenotype [368]. Figure 5 lists non-muscle biofluid markers that can be useful in screening dystrophinopathy-associated changes in the liver (glutamate dehydrogenase isoform GLUD1 and haptoglobin) and the heart (cardiac troponin subunit cTnI), as well as in carrying out the functional evaluation of the kidneys (cystatin-C/CST3 and uromodulin). Measurement of the serum carbonic anhydrase isoform CA3 alone versus the ratio of serum myoglobin to CA3 can be used to determine the loss of integrity of dystrophic skeletal muscles versus dystrophin-deficient heart muscle [423]. An interesting new source of non-invasive biomarkers is represented by the saliva proteome [412], whose analysis showed elevated levels of the kallikrein isoform Klk1 in the *mdx-4cv* model of DMD [344,433]. How future multi-omics research initiatives can build on these proteomic findings and establish a more comprehensive pathophysiological picture of dystrophinopathy is briefly outlined in the below section.

### 4.5. Integromics: Systems Biological Multi-Omics Analysis of Dystrophinopathy

For the future advancement of precision medicine and individualized patient treatments, the implementation of multi-omics biomarker signatures would be an advantage for the detailed evaluation of complex human disease initiation and progression [434]. Ideally, a wide range of multi-modal omics markers at the levels of the genome, transcriptome, epigenome, proteome, metabolome and cytome would be used in combination to decisively enhance the accuracy of diagnostic, prognostic and therapeutic monitoring procedures [435]. The cytome can be defined as the entire collection of dynamic cellular processes, incorporating both structural and functional parameters, that form the basis of all biochemical and physiological processes in the body [436,437,438]. In the field of neurological and neuromuscular disorders, this would be a strategic step forward to evolve evidence-based medicine to the next level of stratified approaches and establish personalized medical therapies via translational neuroscience [439].

In relation to studying the downstream effects of dystrophin deficiency, a holistic systems biological analysis of the complex pathogenesis of DMD would greatly enhance our insights into organ crosstalk and encompass an integrative multi-omics approach. This integromics strategy would ideally consist of genomics [440,441,442], transcriptomics [443,444,445], top-down proteomics [17,59,60,61], bottom-up proteomics [46,107], subproteomics [108,109,110,213], the proteomic evaluation of PTMs [117,291,446,447], metabolomics [448,449], lipidomics [450,451,452], glycomics [453], immunomics [454], secretomics [218,219,220,221,455] and high-throughput cytomics [158,172,456,457], as summarized in Figure 6.

Importantly, the integration of proteomic data sets from both bottom-up and top-down approaches would be extremely helpful in identifying and characterizing specific proteoforms in highly complex tissue systems [458,459]. This could especially help us better understand the enormous complexity of aberrant cellular signaling events that occur in DMD [460]. The wider application of spatial single-myofiber MS analysis would be extremely helpful with protein biochemical studies using techniques such as deep visual proteomics [461]. This method integrates high-content imaging with laser micro-dissection and multiplexed MS-based analyses at the single-cell level and has therefore been termed ‘single-cell Deep Visual Proteomics’ (scDVP) [462]. Besides metagenomics, an additional field of interest for studying the genome is epigenomics [463], which focuses on the systematic analysis of molecular modifications at the level of DNA that may alter the regulation of gene activity but are mitotically stable and are independent of the DNA sequence [464,465,466]. The determination of histone modifications, DNA methylation and the generation of modulatory non-coding RNAs can generate crucial proteogenomic data with considerable relevance to skeletal muscle development, repair and physiological functioning [467,468,469] and the treatment of muscular dystrophy [470,471,472].

Multi-omics approaches have a great potential to improve our understanding of complex human disease mechanisms [473] and establish systems biological concepts [474], including the systems biology of skeletal muscles [475]. The application of multi-omics has already been used to study crucial aspects of skeletal muscle cell biology in health and disease [476,477,478,479] and been applied to certain aspects of the field of dystrophinopathy research, including the integrative screening of dystrophic animal models [277,480,481,482], the evaluation of immune responses in muscular dystrophy [483], myogenic remodeling by human pluripotent stem cells [484], astrocyte-related abnormalities [485] and dystrophinopathy-associated cardiomyopathy [324,331]. The main techniques used for proteomics-centric and multi-omics studies have been recently reviewed by Rajczewski et al. [486]. Combining findings from these types of molecular and cellular analyses will be extremely helpful in predicting the trajectory of disease progression in clinical subtypes of DMD and determining the potential influence of environmental factors and lifestyle. To achieve a maximum yield of data from integrative analyses [35], multi-omics approaches will be assisted by employing big data analytic tools and using optimized machine learning and artificial intelligence approaches [487,488,489,490,491]. A crucial aspect of the development of novel therapies to treat dystrophinopathies is the elucidation of the underlying mechanisms that generate mild forms of DMD [492] or naturally protective phenotypes, such as the spared muscular systems of the tongue, intrinsic laryngeal muscles and extraocular muscles [196,493,494]. Multi-omics analyses of these specialized skeletal muscles that are only mildly affected, despite the fact of dystrophin deficiency, might identify novel targets for therapeutic intervention.

## 5. Conclusions

In this perspective article, we have attempted to address the question of how MS-based proteomic analyses can be employed to better comprehend the pathophysiological complexity and multi-systems involvement of DMD. Integrating findings from top-down proteomics, bottom-up proteomics and subproteomics could be used to establish a more precise picture of the molecular and cellular pathogenesis of this monogenetic disease of the neuromuscular system. Current limitations of MS-based studies of skeletal muscles are based on the fact that only the technically accessible proteome can be studied. Thus, the expansion of the measurable part of the total muscle proteome through improved tissue extraction and protein separation methodology would decisively widen the scope of muscular dystrophy research. Importantly, proteomics represents an essential method of modern systems biology, which is central to the multi-modal omics analysis of complex pathobiochemical mechanisms. The integration of data generation from traditional biological disciplines, such as histochemistry, physiology, biochemistry and molecular biology, with findings from high-throughput and large-scale bioanalytical approaches, including genomics, transcriptomics, proteomics and metabolomics, promises a genuine systems biological understanding of dystrophinopathies. In the future, multi-omics will be instrumental for a more detailed determination of the pathogenic mechanisms and multi-system dysfunction due to dystrophin deficiency. The establishment of a broad and reliable biomarker signature at all levels of biological organization, ranging from the genome to the physiome, will improve screening procedures, differential diagnostics and prognostic predictions, as well as expand the systematic identification of new therapeutic targets to treat dystrophinopathy. In the long term, if integrated proteomics and multi-omics approaches are properly established for studying detailed mechanisms of DMD and biomarkers are verified for their clinical suitability, this biomedical information can be consolidated to have a considerable clinical impact.

## Figures and Tables

**Figure 1 proteomes-12-00004-f001:**
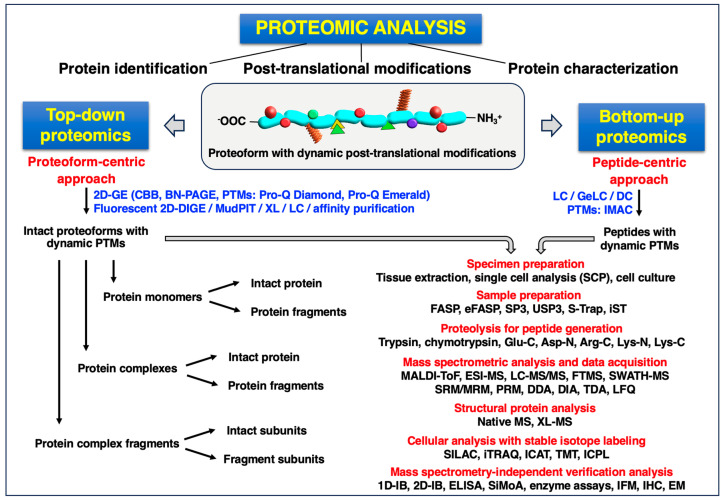
Outline of the main proteomic approaches that are routinely utilized to isolate, identify and characterize proteins and their individual proteoforms. Abbreviations used: 1D, one-dimensional; 2D, two-dimensional; BN-PAGE, blue native polyacrylamide gel electrophoresis; CBB, Coomassie Brilliant Blue; DC, differential centrifugation; DDA, data-dependent acquisition; DIA, data-independent acquisition; DIGE, difference gel electrophoresis; eFASP, enhanced filter-aided sample preparation; ELISA, enzyme-linked immunosorbent assay; EM, electron microscopy; ESI, electrospray ionization; FASP, filter-aided sample preparation; FT, Fourier-transform ion cyclotron resonance; GE, gel electrophoresis; GeLC, gel electrophoresis–liquid chromatography; IB, immunoblotting; ICAT, isotope-coded affinity tags; ICPL, isotope-coded protein labeling; IFM, immunofluorescence microscopy; IHC, immunohistochemistry; iST, In-StageTip; LC, liquid chromatography; IMAC, immobilized metal affinity chromatography; iTRAQ, isobaric tagging for relative and absolute quantitation; LFQ, label-free quantification; MALDI, matrix-assisted laser desorption/ionization; MS, mass spectrometry; MudPIT; multi-dimensional protein identification technology; PRM, parallel reaction monitoring; PTMs, post-translational modifications; SCP, single-cell proteomics; SILAC, stable isotope labeling by amino acids in cell culture; SiMoA, Simoa bead-based immunoassay; SP3, single-pot solid-phase-enhanced sample preparation; SRM/MRM, selected/multiple reaction monitoring; S-Trap, suspension trapping; SWATH, Sequential Window Acquisition of all Theoretical Mass Spectra; TDA, targeted data acquisition; TMT, isobaric tandem mass tagging; ToF, time-of-flight; USP3, universal solid-phase protein preparation; XL, cross-linking.

**Figure 2 proteomes-12-00004-f002:**
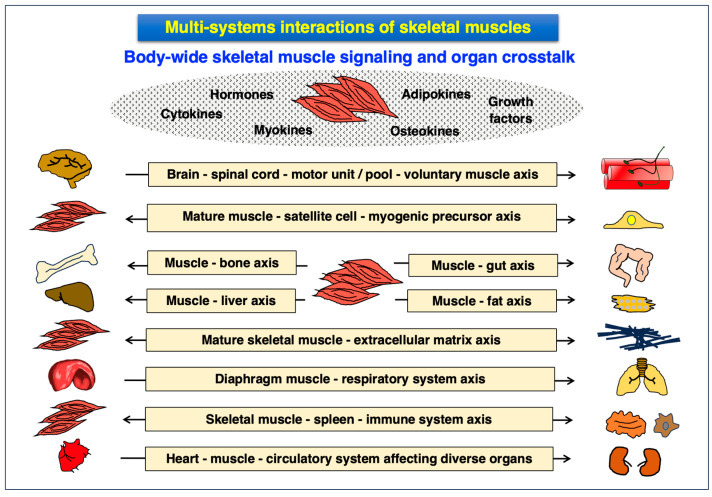
Body-wide signaling axes that involve the neuromuscular system. Shown are major patterns of organ crosstalk that involve voluntary muscles, the central nervous system, bone, liver, kidneys, the gastrointestinal tract, fatty tissue, the immune system, the lungs and the heart. At the level of the neuromuscular system, intensive signaling events occur between matured myofibers and their innervating motoneuron, as well as between contractile fibers and their environment consisting of the extracellular matrix and stem cells that can be activated to form myogenic precursors during repair processes. Major signaling factors that are involved in muscle adaptations, myofiber repair, muscle–bone interactions, activation of the muscle-associated immune response, metabolic regulation and bioenergetic processes include myokines, which constitute the dynamic skeletal muscle secretome, as well as a variety of hormones, growth factors, cytokines, osteokines and adipokines.

**Figure 3 proteomes-12-00004-f003:**
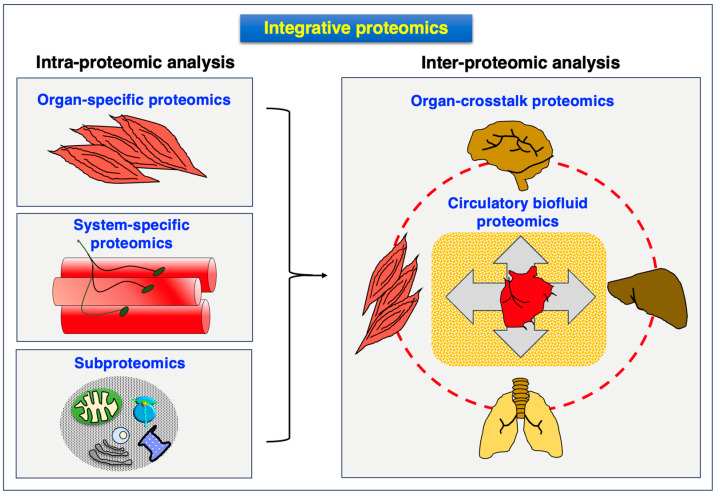
Overview of the main approaches used in integrative proteomics. Listed are organ-specific proteomics that focuses on the analysis of individual skeletal muscles, system-specific proteomics for the in-depth analysis of the nerve–muscle connection and motor units, and subproteomics that centers on the mass spectrometric characterization of distinct organelles and supramolecular protein assemblies. Organ-crosstalk proteomics is concerned with the analysis of the circulatory biofluid proteome and how the release of myokines affects other organ systems in the body.

**Figure 4 proteomes-12-00004-f004:**
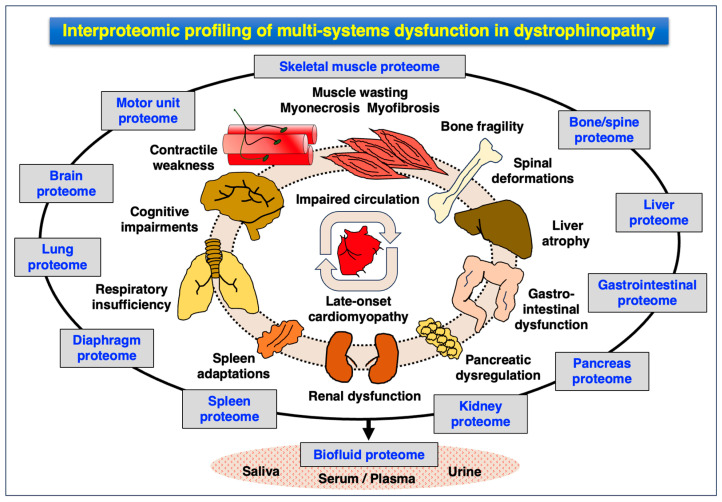
The pathoproteomic profile of multi-system changes in Duchenne muscular dystrophy. The diagram outlines the complexity of body-wide alterations due to dystrophin deficiency and illustrates how the systematic application of a comprehensive interproteomic profiling approach could help us better understand the multi-system dysfunction in dystrophinopathy.

**Figure 5 proteomes-12-00004-f005:**
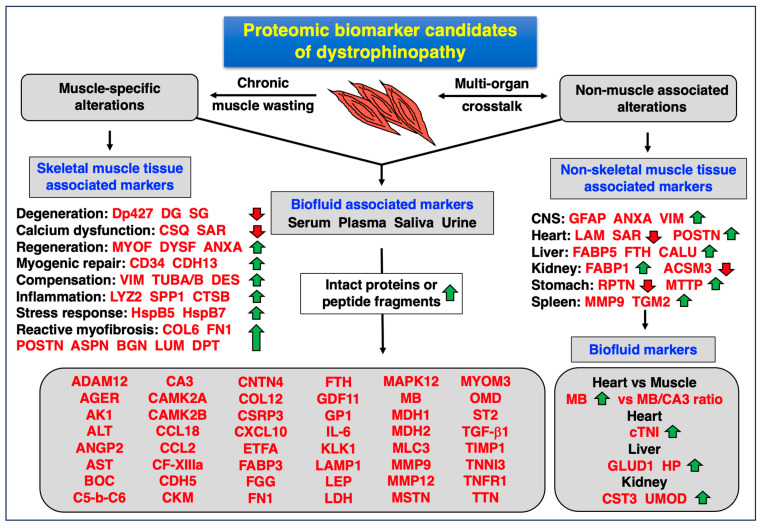
Overview of biomarker candidates of dystrophinopathy as determined by biochemical screening and mass spectrometry-based proteomics. Listed are both tissue-associated changes and biofluid-related alterations in specific protein species. Skeletal muscle tissue markers are categorized according to their involvement in myofiber degeneration, abnormal calcium handling, regeneration and repair mechanisms, compensatory processes, chronic inflammation and macrophage invasion, the cellular stress response and reactive myofibrosis. Individual muscle-derived biofluid markers were identified as intact proteins or fragments, as in the case of the giant protein titin. Non-skeletal muscle markers are listed for proteomic changes in the brain, heart, liver, kidney, stomach and spleen. Non-muscle biofluid markers are described for the differential analysis of changes in skeletal muscle versus the heart, the analysis of the liver and functional evaluation of the kidneys. Increases versus decreases in biomarkers are marked by green and red arrows, respectively. Abbreviations used: ACSM, acyl-CoA synthetase medium chain; ADAM, disintegrin and metalloproteinase domain-containing protein; AGER, advanced glycosylation end-product specific receptor; AK, adenylate kinase; ALT, alanine aminotransaminase; ANGP, angiopoietin; ASPN, asporin; AST, aspartate transaminase; ANXA, annexin A; BNG, biglycan; BOC, brother of CDO (CAM-related/down-regulated by oncogenes); C5-b-C6, complement components; CA, carbonic anhydrase; CALU, calumenin; CAMK, Ca^2+^/calmodulin-dependent protein kinase; CCL, C-C motif chemokine ligand; CD34, hematopoietic progenitor cell antigen CD34; CF, coagulation factor; CDH, cadherin; CKM, creatine kinase, muscle type; CNS, central nervous system; GLUD1, glutamate dehydrogenase; CNTN, contactin; COL, collagen; CSQ, calsequestrin; CSRP, cysteine- and glycine-rich protein; CST, cystatin; cTNI, cardiac troponin I; CTSB, cathepsin B; CXCL, C-X-C motif chemokine ligand; DES, desmin; DG, dystroglycan; Dp427, dystrophin; DPT, dermatopontin; DYSF, dysferlin; ETFA, electron transfer flavoprotein subunit alpha; FABP, fatty acid-binding protein; FGG, fibrinogen gamma chain; FN, fibronectin; FTH, ferritin heavy chain; GDF, growth differentiation factor; GFAP, glial fibrillary acidic protein; GP, glycoprotein; HP, haptoglobin; Hsp, heat shock protein; IL, interleukin; KLK, kallikrein; LAM, laminin; LAMP, lysosomal-associated membrane protein; LEP, leptin; LDH, lactate dehydrogenase; LUM, lumican; LYZ, lysozyme; MAPK, mitogen-activated protein kinase; MB, myoglobin; MDH, malate dehydrogenase; MLC, myosin light chain; MMP, matrix metallo-proteinase; MSTN, myoststin; MTTP, microsomal triglyceride transfer protein; MYOM, myomesin; MYOF, myoferlin; OMD, osteomodulin; POSTN, periostin; RPTN, repetin; SAR, sarcalumenin; SG, sarcoglycan; SPP, secreted phosphoprotein; ST, suppression of tumorigenicity; TGF, transforming growth factor; TGM, transglutaminase; TIMP, tissue inhibitor of metalloproteinase; TNNI, troponin subunit I; TNFR, tumor necrosis factor receptor; TTN, titin; TUBA/B, tubulin chains A and B; UMOD, uromodulin; VIM, vimentin.

**Figure 6 proteomes-12-00004-f006:**
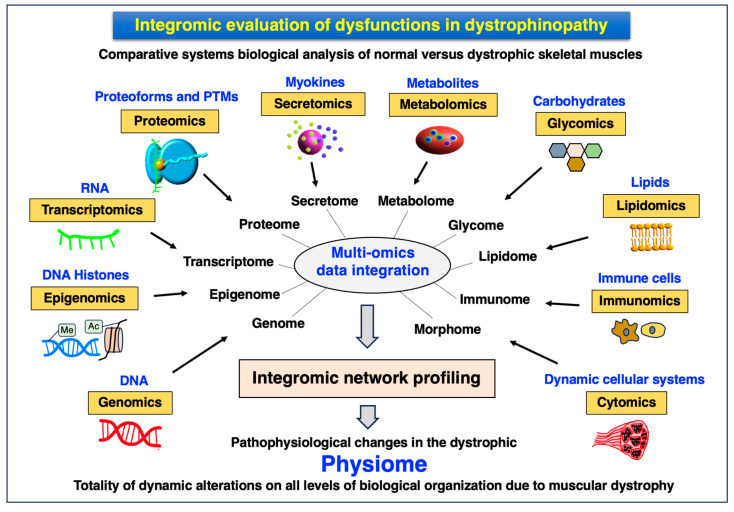
Integromic analysis of dysfunctions in Duchenne muscular dystrophy. Multi-omics-based investigations are envisaged to generate a more comprehensive understanding of the molecular and cellular complexity of the pathogenic mechanisms that are involved in dystrophinopathy.

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
