# Peer review of "How Can Proteomics Help to Elucidate the Pathophysiological Crosstalk in Muscular Dystrophy and Associated Multi-System Dysfunction?"

_proteomes, 2024, doi:10.3390/proteomes12010004_

Round 1

Reviewer 1 Report

Comments and Suggestions for Authors

This perspective offers a valuable and timely exploration of utilizing proteomics to uncover molecular mechanisms and multi-system pathology in Duchenne muscular dystrophy (DMD). The authors comprehensively summarize key findings from skeletal muscle and non-muscle proteomics in DMD, and make a compelling case for integrating proteomics into a multi-omics "integromics" approach to gain systems-level insights. The graphical summaries are clear and helpful for visualizing complex concepts. The perspective is well-written, highly informative and discusses implications for advancing biomarker development and precision medicine in DMD. I recommend this insightful article for publication following minor revisions.

Specific comments:

1.       Introduction - Concisely explains the rationale and context very well. The only addition I would suggest is briefly mentioning the potential of proteomics for identifying therapeutic targets or drug mechanisms early on.

2.       Section 4.4 Biomarkers - The overview figure integrating biomarker candidates is outstanding. In the text, highlight 1-2 of the most promising serum, urine or saliva biomarkers.

3.       Conclusion - Succinctly encapsulates key points. Consider adding a statement about potential clinical impact if integrated proteomics and multi-omics can yield enough insights and biomarkers.

4.       The conclusion section (number 6) should be renumbered as 5 for organizational consistency.

Author Response

Reviewer 1, Comment 1: ‘This perspective offers a valuable and timely exploration of utilizing proteomics to uncover molecular mechanisms and multi-system pathology in Duchenne muscular dystrophy (DMD). The authors comprehensively summarize key findings from skeletal muscle and non-muscle proteomics in DMD, and make a compelling case for integrating proteomics into a multi-omics "integromics" approach to gain systems-level insights. The graphical summaries are clear and helpful for visualizing complex concepts. The perspective is well-written, highly informative and discusses implications for advancing biomarker development and precision medicine in DMD. I recommend this insightful article for publication following minor revisions’.

Response: We would like to thank Reviewer 1 for the positive evaluation of our manuscript. As suggested, we have carried out revisions of our manuscript as outlined in detail in below responses to Reviewer 1. Both, an unmarked and revised version and a manuscript with highlighted changes have been uploaded. The below mentioned page numbers correlate to the revised R1 version with highlighted changes.

Reviewer 1, Comment 2: ‘Specific comments: 1. Introduction - Concisely explains the rationale and context very well. The only addition I would suggest is briefly mentioning the potential of proteomics for identifying therapeutic targets or drug mechanisms early on’.

Response: We agree and have revised the Introduction section accordingly and have added a statement that proteomics is also highly suitable for the identification of novel therapeutic targets and the elucidation of drug mechanisms.

Revised Page 2: ‘… of bioanalytical advantages versus technical challenges, is provided. Besides studying the molecular pathogenesis of dystrophinopathy and being an irreplicable tool for biomarker discovery, proteomics is also highly suitable for the identification of novel therapeutic targets and the elucidation of drug mechanisms [34]. Based on the high-throughput and …’.

Reviewer 1, Comment 3: ‘Specific comments: 2. Section 4.4 Biomarkers - The overview figure integrating biomarker candidates is outstanding. In the text, highlight 1-2 of the most promising serum, urine or saliva biomarkers’.

Response: We agree and have revised Section 4.4 and have added this information, as follows:.

Revised Page 19: ‘… and massive fibrosis of dystrophic muscles [422]. In our opinion, the most suitable minimally invasive and biofluid-associated biomarkers of DMD are represented by amino-terminal and carboxy-terminal TTN fragments of titin in urine, kallikrein KLK1 in saliva and the combination of CA3/FABP3/MYOM3/MDH2 in plasma/serum samples.

Reviewer 1, Comment 4: ‘Specific comments: 3. Conclusion - Succinctly encapsulates key points. Consider adding a statement about potential clinical impact if integrated proteomics and multi-omics can yield enough insights and biomarkers’.

Response: To address this point, we have revised the Conclusions section and have added the following statement:

Revised Page 19: ‘… targets to treat dystrophinopathy. In the long-term, if integrated proteomics and multi-omics approaches are properly established for studying detailed mechanisms of DMD and biomarkers are verified for their clinical suitability, this biomedical information can be consolidated to have a considerable clinical impact’.

Reviewer 1, Comment 5: ‘Specific comments: 4. The conclusion section (number 6) should be renumbered as 5 for organizational consistency’.

Response: We would like to thank Reviewer 1 to point out this inconsistency in the numbering of sections and have corrected this, as follows:

Revised Page 23: ‘… 5. Conclusions.  In this perspective article, we have …’.

We would like to thank Reviewer 1 for the constructive criticism of our manuscript and hope that you will find the revised version of our article now acceptable for publication in Proteomes.

Reviewer 2 Report

Comments and Suggestions for Authors

In their review, Dowling et al. address how proteomics might increase our understanding of the molecular pathogenesis of complex diseases like muscular dystrophy. The authors propose a holistic and integromic bioanalysis that would integrate diverse omics-type studies including inter- and intra-proteomics with modern biomolecular analyses. Overall, this is a relatively well-written review article that provides an in-depth discussion of modern proteomics methods and their application to the study of muscular dystrophy. However, there are some major and minor issues that the authors I would like the authors to tackle prior to my being willing to recommend their manuscript for publication. These issues are briefly outlined below.

Major Issues:

1)    My biggest issue with this review manuscript is that it spends 9.5 pages defining mass spectrometry approaches, analysis, and data acquisition techniques. Consequently, it reads more as a chapter from a textbook on proteomics than a perspective article. Is there any way that the authors could simplify these first 9.5 pages or work them into the remaining 13.5 pages of their review?

2)    My second major issue with this work is that it is highly repetitive. For example, the authors describe in detail various mass spectrometry approaches in the text of their article and then they repeat this information in their manuscript’s figures, tables, and lists. Ultimately, this increases the length of their manuscript unnecessarily. 

3)    In Figure 2 and various points throughout the text, the authors discuss several “Body-wide signaling axes”; however, these “axes” are poorly defined and rather nebulous. 

4)    Figure 3 discusses “inter-proteomic analysis” in such a way that suggests that all non-muscle phenotypes observed in muscular dystrophy are the result of “organ-crosstalk” via the circulatory system. However, it could be that these non-muscle phenotypes may result instead from tissue-specific effects of genetic mutations. Perhaps the authors could consider both tissue autonomous vs. non-autonomous hypotheses in their review?

5)    Figure 4 is similar to Figure 3, in that it appears that all “multi-systems changes in Duchenne muscular dystrophy” result from changes in the biofluid proteome. The arrowheads present in this figure lead me to this conclusion.

6)    Figure 5 is similar to Figures 3 and 4. Is it always “multi-organ crosstalk” or tissue-specific defects caused by genetic mutations-associated with dystropinopathy? In addition, it would be helpful if the authors could indicate, potentially via color, if a particular biomarker is up- or down-regulated in dystrophinopathy.

Minor Issues:

1)    The authors should define the term “proteoform” for their readers who may not be familiar. This definition should be provided upon the first use of “proteoform” in the manuscript (Page 2, Line 49).

2)    Page 2, Line 86: The title of this section should also include reference to “middle-up/down proteomics”, as they are discussed in addition to “top-down” and “bottom-up” approaches. 

3)    Page 2, Lines 126-127: The word “perspectives” should be replaced with “perspective”.

4)    Page 5, Line 186: The word “proteins” should be replaced with “protein”.

5)    Page 7, Line 290: The term “MS2 spectra” should be defined for readers who may not be familiar.

6)    Page 10, Line 275: The word “myosin” should be replaced with “myosin II”.

7)    Page 10, Line 407: What exactly do the authors mean by “physically tough” here? Difficult to make lysates from? Similarly, on Page 12, Line 454 the use of “tough types” is confusing.

8)    Page 11, Line 412: The term “myokine” needs to be briefly defined for readers who may not be familiar with it.

9)    Page 13, Lines 498-501: What do the authors mean by “were established to trigger” here? Established in genetically engineered mouse models?

10) Page 13, Line 520: It is unclear what the authors mean by “dystrophin node” here.

11) Page 14, Line 547: Which “lamin” are the authors referring to here? A-type lamins or B-type lamins, or both?

12) Page 16, Line 638: What do the authors mean by “A solid immune response” here?

13) Page 19, Line 778: Which “tubulin” are the authors referring to here? There are multiple tubulin genes. This is also an issue in Figure 5’, with “TUB”.

Comments on the Quality of English Language

Overall, the quality of English used in this article is fine. 

Author Response

Reviewer 2, Comment 1: ‘In their review, Dowling et al. address how proteomics might increase our understanding of the molecular pathogenesis of complex diseases like muscular dystrophy. The authors propose a holistic and integromic bioanalysis that would integrate diverse omics-type studies including inter- and intra-proteomics with modern biomolecular analyses. Overall, this is a relatively well-written review article that provides an in-depth discussion of modern proteomics methods and their application to the study of muscular dystrophy. However, there are some major and minor issues that the authors I would like the authors to tackle prior to my being willing to recommend their manuscript for publication. These issues are briefly outlined below’.

Response: We would like to thank Reviewer 2 for the critical evaluation of our manuscript. As suggested, we have carried out revisions of our manuscript as outlined in detail in below responses to Reviewer 2. Both, an unmarked and revised version and a manuscript with highlighted changes have been uploaded. The below mentioned page numbers correlate to the revised R1 version with highlighted changes.

Reviewer 2, Comment 2: ‘Major Issues: 1) My biggest issue with this review manuscript is that it spends 9.5 pages defining mass spectrometry approaches, analysis, and data acquisition techniques. Consequently, it reads more as a chapter from a textbook on proteomics than a perspective article. Is there any way that the authors could simplify these first 9.5 pages or work them into the remaining 13.5 pages of their review?’.

Response: We’d like to thank Reviewer 2 for the critical assessment of our manuscript and respect the opinion on the extensive outlining of MS methods. To address the reviewer’s concern, we have shortened substantial parts in revised Section 2, as outlined below. However, we believe that Section 2 is essential for properly setting the scene on how proteomic methodology can be applied in modern biochemical research. Proteomics is a technology-driven approach and as such the techniques used are of central importance for understanding the proteomic perspective of muscular dystrophy research. In our opinion, Perspective papers should give experts in the field the opportunity to freely design and structure their paper about a particular scientific topic and/or analytical approach. Hence, editors and referees should ideally review Perspective articles for quality and relevance of argument only, and not necessarily agree with the author’s concepts about structuring of their paper and scientific directions.

The following parts of Section 2 have been removed and are highlighted in GREEN in the marked R1 version:

Revised Page 4: REMOVED: ‘In the first dimension, protein molecules are resolved depending on their pI-value, with a wide-range of immobilized pH-gradient (IPG) strips being available, including strips with a wide-range pH-gradient (pH 3-11) or different narrow-range pH gradients (such as pH 4-7) to optimize the separation of specific proteins of interest [64]. IPG strips are available in different lengths, e.g. 7 and 18 cm, to cater for different size resolving gels that are to be included in the experiment. In the second dimension, protein separation is performed based on molecular weight using SDS Laemmli-type or Tris-Tricine buffer systems [64]. The typical size of resolving gels includes 8x7cm2, 23x30cm2 and 40x30cm2. The specific usage of particular gels depends on the sample characteristics and the degree of resolution needed’.

Revised Page 4: REMOVED: ‘2D-GE can be effectively combined with other biochemical techniques, facilitating gel staining, followed by spot excision, de-staining, protein digestion, and the subsequent analysis of peptides by MS analysis to unequivocally identify specific proteoforms’.

Revised Page 4: REMOVED: ‘Visualized protein spots can be excised manually or with an automatic spot picker for digestion and subsequent MS-based analysis’.

Revised Page 5: REMOVED: ‘Common gel stains such as SYPRO Ruby, silver stain and CBB can be used for quantitative protein detection in 2D-gels, but the number of gels needed for a comparative analysis and the compressed dynamic range make this approach relatively re-strictive’.

Revised Page 5: REMOVED: ‘When starting material is scarce, … is the method of choice for quantitative proteome analysis,’.

Revised Page 5: REMOVED: ‘This widely used approach ultimately detects and measures peptides, which represent ideal molecular species for the swift MS-based proteomic analysis due to the fact that they readily solubilize, separate and ionize’.

Revised Page 5: REMOVED: ‘… (gel filtration/size exclusion - resins are porous to molecules with a particular size range), … (ion exchange – anion/cation), … (a specific binding affinity - 6xHis tag)’.

Revised Page 5: REMOVED: ‘The efficient digestion of proteins into peptides is a critical step in preparing samples for MS analysis [81], as already outlined above in relation to top-down proteomics [82-85]’.

Revised Page 6: REMOVED: ‘Trypsin is highly suitable for digesting proteins into small-size peptides [81], which are more amenable to high-performance liquid chromatography (HPLC) separation and tandem MS (MS/MS) characterization. Alternative enzymes are available to be used alone or in combination with trypsin for the optimum generation of peptide populations prior to MS-based analysis [82-85]. Once peptides have been generated after digestion, silica based octadecyl (C18) resins can be employed to purify and concentrate established peptides. At this stage, …’.

Revised Page 6: REMOVED: ‘Of note, the use of first dimension methods (e.g. size exclusion chromatography/SEC or ion exchange chromatography/IEX) coupled to MS-compatible second dimension approaches (e.g. reversed phase liquid chromatography/RPLC or hydrophilic interaction liquid chromatography/HILIC) and related techniques, greatly improves the resolving power and thereby decisively increases the number of peptides that can be analyzed per run …’.

Revised Page 6: REMOVED: ‘However, ion intensities are more accurate than spectral counts and have a greater dynamic range. A drawback of the LFQ approach is that run parameters, e.g. C18 column conditions, may change marginally between samples, having potentially knock-on consequences with respect to sample alignment and analysis. This limitation is not a significant factor in labelled experiments, as sample comparison is achieved within a single MS run in most cases’.

Revised Page 6: REMOVED: ‘The current version of TMT reagents allows the simultaneous quantification of up to 18 samples (TMTpro 18-plex), through amide bond formation with the amino groups of peptide N-termini and at lysine residues in digested protein mixtures’.

Revised Page 7: REMOVED: ‘A combination of both CID and ETD can overcome some of the limitations when using these approaches individually. CID fragmentation generates spectra with limited peptide backbone fragmentation (b and y ions), whereas ETD creates a more complex series of ions (c and z ions), with the added benefit of leaving labile PTMs intact. HCD is a CID tech-nique that is specific to some classes of MS instruments. The HCD approach is particularly valid when reporter ion interpretation is needed for isobaric tag-based quantification’.

Revised Page 7: REMOVED: ‘Thereafter, many commercially available or open source software packages can be used to manage raw MS data and generate distinct data sets on peptide/protein abundances, employing bespoke algorithms depending on the initially used proteomic workflow’.

Reviewer 2, Comment 3: ‘Major Issues: 2) My second major issue with this work is that it is highly repetitive. For example, the authors describe in detail various mass spectrometry approaches in the text of their article and then they repeat this information in their manuscript’s figures, tables, and lists. Ultimately, this increases the length of their manuscript unnecessarily’. 

Response: In contrast to the opinion of Reviewer 2, in previous invited articles in Proteomes and other MDPI journals, we were strongly encouraged by both editors and reviewers to use illustrative figures and tables to strength the main points made in a review/perspective article. We agree with this publishing approach, especially in the case of Reviews/Perspective articles, and believe that figures and tables are highly suitable to summarize and underline important aspects within a Perspective paper. In our opinion, repetition in text and figures does not diminish the impact of a paper, but instead can be the main strength of an article that reviews a particular scientific aspect.

Reviewer 2, Comment 4: ‘Major Issues: 3) In Figure 2 and various points throughout the text, the authors discuss several “Body-wide signaling axes”; however, these “axes” are poorly defined and rather nebulous’. 

Response: In our opinion, both the concept of myokine signalling and associated signalling axes between skeletal muscles, which are distributed throughout the body, and other cells/tissues/organs/organ systems are relatively well established in the field of basic and applied myology. Several reviews are available that outline in detail both the cell biological concept and scientific evidence for muscle-associated signalling axes. Please see: Severinsen, M.C.K.; Pedersen, B.K. Muscle-Organ Crosstalk: The Emerging Roles of Myokines. Endocr. Rev. 2020, 41, 594–609. https://doi.org/10.1210/endrev/bnaa016; and Kirk, B.; Feehan, J.; Lombardi, G.; Duque, G. Muscle, Bone, and Fat Crosstalk: the Biological Role of Myokines, Osteokines, and Adipokines. Curr. Osteoporos. Rep. 2020, 18, 388–400. https://doi.org/10.1007/s11914-020-00599-y; and Lara-Castillo, N.; Johnson, M.L. Bone-Muscle Mutual Interactions. Curr. Osteoporos. Rep. 2020, 18, 408–421. https://doi.org/10.1007/s11914-020-00602-6; and Gomarasca, M.; Banfi, G.; Lombardi, G. Myokines: The endocrine coupling of skeletal muscle and bone. Adv. Clin. Chem. 2020, 94, 155–218. https://doi.org/10.1016/bs.acc.2019.07.010. We believe that these reviews strongly suggest that body-wide interaction patterns exist at the level of myokine signalling, and these papers have been quoted in our article to support the idea of organ crosstalk with skeletal muscles.

Reviewer 2, Comment 5: ‘Major Issues: 4) Figure 3 discusses “inter-proteomic analysis” in such a way that suggests that all non-muscle phenotypes observed in muscular dystrophy are the result of “organ-crosstalk” via the circulatory system. However, it could be that these non-muscle phenotypes may result instead from tissue-specific effects of genetic mutations. Perhaps the authors could consider both tissue autonomous vs. non-autonomous hypotheses in their review?’.

Response: We like to thank Reviewer 2 for this important point. We agree that the tissue-specific expression of the various dystrophins can be differentially affected due to various mutations in the DMD gene. Thus, not all effects are due to organ-crosstalk, but can be based on mutation-specific alterations in non-muscle tissues. This point has now been stated in the revised manuscript.

Revised Page 13: ‘… It is important to stress that changes in non-muscle phenotypes in muscular dystrophies are most likely due to a combination of both organ-crosstalk via the circulatory system and intrinsic changes within individual tissues/organs. This is especially relevant to DMD where it is known that the DMD gene contains several promoters that produce 8 different tissue-specific dystrophin isoforms, as outlined in below section on the genetic basis of dystrophinopathy. The tissue-specific expression of dystrophins can be differentially affected by various mutations in the DMD gene. Thus, not all non-muscle effects are due to organ-crosstalk, but can be based on mutation-specific alterations in non-muscle tissues’.

Reviewer 2, Comment 6: ‘Major Issues: 5) Figure 4 is similar to Figure 3, in that it appears that all “multi-systems changes in Duchenne muscular dystrophy” result from changes in the biofluid proteome. The arrowheads present in this figure lead me to this conclusion’.

Response: To address this potentially confusing issue, we have removed the arrowheads in revised Figure 4. In contrast to Figure 3, which gives a general overview of the main approaches used in integrative proteomics, Figure 4 focuses on the pathoproteomic profile of multi-systems changes in muscular dystrophy. This gives the illustrations of these two figures very different messages, one on the general concept of integrative proteomics and the other on the complexity of body-wide alterations due to dystrophin deficiency and how the systematic application of a comprehensive interproteomic profiling approach might help to better understand the multi-systems dysfunction in dystrophinopathy.

Reviewer 2, Comment 7: ‘Major Issues: 6) Figure 5 is similar to Figures 3 and 4. Is it always “multi-organ crosstalk” or tissue-specific defects caused by genetic mutations-associated with dystropinopathy? In addition, it would be helpful if the authors could indicate, potentially via color, if a particular biomarker is up- or down-regulated in dystrophinopathy’.

Response: As already discussed above, we agree on the point of mutation-associated changes in non-muscle tissues. This is now stated on revised Page 13. In our opinion, Figure 5 is not a repetition of Figures 3 and 4. This figure provides a list of proteomic biomarker candidates related to dystrophinopathy. As suggested by Reviewer 2, changes in biomarkers are now better marked by arrows in red and green colour.

Revised Figure 5: ‘Figure 5. Overview of biomarker candidates of dystrophinopathy as determined by biochemical screening and mass spectrometry-based proteomics. ………. and functional evaluation of the kidneys. Increases versus decreases in biomarkers are marked by red and green arrows, respectively. Abbreviations used: …….; TUBA/B, tubulin chains A and B; …’.

Reviewer 2, Comment 8: ‘Minor Issues: 1) The authors should define the term “proteoform” for their readers who may not be familiar. This definition should be provided upon the first use of “proteoform” in the manuscript (Page 2, Line 49)’.

Response: The revised manuscript now defines the term ‘proteoform’.

Revised Page 2: ‘… Proteoforms can be defined as the expressed variants of the protein products that are encoded by a single gene, whereby the different molecular forms are generated by genetic variations such as alternative promoter usage, alternative splicing of RNA transcripts due to mechanisms such as exon skipping, and extensive post-translational modifications, including proteolysis, phosphorylation and glycosylation [12].’

Reviewer 2, Comment 9: ‘Minor Issues: 2) Page 2, Line 86: The title of this section should also include reference to “middle-up/down proteomics”, as they are discussed in addition to “top-down” and “bottom-up” approaches’.

Response: As suggested, the revised manuscript now also uses the term ‘middle-up/down’ in the title of Section 2.

Revised Page 2: ‘2. Mass spectrometry-based proteomics: top-down versus middle-up/down versus bottom-up approaches’.

Reviewer 2, Comment 10: ‘Minor Issues: 3) Page 2, Lines 126-127: The word “perspectives” should be replaced with “perspective”.’.

Response: This point has been addressed in the revised manuscript.

Revised Page 4: ‘Detailed descriptions of the key techniques employed in MS-based proteomics are beyond the scope of this perspective article that instead focuses on the actual application of proteomics for a more in-depth understand of the pathobiochemical aspects of the multi-systems pathology of dystrophinopathy’.

Reviewer 2, Comment 11: ‘Minor Issues: 4) Page 5, Line 186: The word “proteins” should be replaced with “protein”.’.

Response: This point has been addressed in the revised manuscript.

Revised Page 5: ‘Typically, the internal control is a combination of all samples that will be analyzed within a single experiment and is labelled with CyDye2. The CyDye3 and CyDye5 labelled samples can then be normalized to CyDye2 for the identification of protein spots with different abundance levels when comparing samples [91-93]’.

Reviewer 2, Comment 12: ‘Minor Issues: 5) Page 7, Line 290: The term “MS2 spectra” should be defined for readers who may not be familiar’.

Response: The revised manuscript now defines the term ‘MS2 spectra’.

Revised Page 7: ‘… and data-dependent acquisition (DDA) [139]. During MS/MS analysis, MS2 spectra are produced from the fragmentation of a product ion of a particular m/z range, following operation of MS1 at scan mode. In DIA, effectively all peptides …’.

Reviewer 2, Comment 13: ‘Minor Issues: 6) Page 10, Line 275: The word “myosin” should be replaced with “myosin II”.’.

Response: This issue was addressed on Page 10 of the revised manuscript.

Revised Page 10: ‘Reliable biochemical/proteomic markers for the different myofiber types are rep-resented by the isoforms of the contractile protein myosin II’.

Reviewer 2, Comment 14: ‘Minor Issues: 7) Page 10, Line 407: What exactly do the authors mean by “physically tough” here? Difficult to make lysates from? Similarly, on Page 12, Line 454 the use of “tough types” is confusing’.

Response: To address this point, we have removed the potentially confusing terms ‘physically tough’ and ‘tough types’ and rearranged these sentences. What was meant is that muscle tissue is not soft, such as liver tissues that can be easily homogenized, but difficult to lyse/homogenize due to the specific nature of muscle tissues, which is characterized by large and elongated myofibers, a high abundance of sarcomeric structures and several layers of extracellular matrix. This makes it more difficult to produce muscle homogenates.

Revised Page 11: ‘… are heterogeneous in composition. Of note, due to the specific nature of muscle tissues, which is characterized by large and elongated myofibers, a high abundance of sarcomeric structures and several layers of extracellular matrix, in combination with associated issues of subcellular fractionation and protein extraction procedures, only a near-to-complete coverage of the skeletal muscle proteome is currently possible [213].’.

Revised Page 12: ‘… muscle proteome [185]. Skeletal muscles contain both tissues that are difficult to homogenize and comprise of a large amount of sarcomeric proteins, membrane-associated proteins and relatively insoluble proteins of the ECM, making the proteomic analysis of total extracts a difficult task. Proteome-wide effects on …’.

Reviewer 2, Comment 15: ‘Minor Issues: 8) Page 11, Line 412: The term “myokine” needs to be briefly defined for readers who may not be familiar with it’.

Response: This term has now be defined on revised Page 11.

Revised Page 11: ‘Myokines can be defined as peptides or proteins that are released or secreted by skeletal muscles into the circulatory system and exert autocrine, paracrine and endocrine effects. Hence, these muscle-derived signaling factors influence the muscle itself at the local level, as well as trigger changes in short- or long-term distant cells/tissues/organs [214]’.

Reviewer 2, Comment 16: ‘Minor Issues: 9) Page 13, Lines 498-501: What do the authors mean by “were established to trigger” here? Established in genetically engineered mouse models?’.

Response: The listed genetic abnormalities relate to the genomic analysis of human mutations in DMD. To address this issue, we substituted ‘established to trigger’ with ‘shown to be associated with’ in this sentence in the revised manuscript.

Revised Page 13: ‘… Diverse types of genetic abnormalities were shown to be associated with dystrophinopathies, including splice site mutations, nonsense point mutations, missense point mutations and mid-intronic mutations, as well as small and large insertions, small and large deletions and large duplications [244-246].’.

Reviewer 2, Comment 17: ‘Minor Issues: 10) Page 13, Line 520: It is unclear what the authors mean by “dystrophin node” here’.

Response: The concept of a ‘dystrophin node’ is an established term in the dystrophin field of research. The dystrophin-containing node of the sarcolemma is defined as the integrating structure of the intracellular cytoskeleton, the central provider of lateral force transmission, the central plasmalemmal hub of fibre stabilisation and a key point for cellular signaling mechanisms in skeletal muscles. The dystrophin node has been previously outlined in detail in an invited review in PROTEOMES. We have added reference [254] behind the term ‘dystrophin node’. Please see Reference [254]: Dowling et al. The Dystrophin Node as Integrator of Cytoskeletal Organization, Lateral Force Transmission, Fiber Stability and Cellular Signaling in Skeletal Muscle. Proteomes. 2021, 9, 9. https://doi.org/10.3390/proteomes9010009.

Revised Page 14: ‘… causes the loss of the organizing dystrophin node [254]. In healthy muscles, the dystrophin-containing node of the sarcolemma is defined as the integrating structure of the intracellular cytoskeleton, the central provider of lateral force transmission, the plasmalemmal hub of fiber stabilization and a key point for cellular signaling mechanisms at the myofiber periphery. In muscular dystrophy, the collapse of the dystrophin node results in …’.

Reviewer 2, Comment 18: ‘Minor Issues: 11) Page 14, Line 547: Which “lamin” are the authors referring to here? A-type lamins or B-type lamins, or both?’.

Response: Proteomic studies have mostly identified changes in ‘B-type lamins’. This information has been added to the revised manuscript.

Revised Page 5: ‘… The most prominent and reproducibly identified proteins include adenylate kinase isoform AK1, annexins, small heat shock proteins, desmin, vimentin, tubulins, collagens, calsequestrin, B-type lamin, myoferlin, dysferlin, ferritin, carbonic anhydrase isoform CA3, fatty acid binding protein FABP3 and various contractile proteins [134,148,180,232,266-292]’.

Reviewer 2, Comment 19: ‘Minor Issues: 12) Page 16, Line 638: What do the authors mean by “A solid immune response” here?’.

Response: To avoid any potential confusion, the term ‘solid’ has been substituted with ‘extensive’ in this sentence.

Revised Page 17: ‘An extensive immune response is observed in X-linked muscular dystrophy causing chronic inflammation in the skeletal musculature [352-354], which presents an excellent opportunity for therapeutic interventions [355-358]’.

Reviewer 2, Comment 20: ‘Minor Issues: 13) Page 19, Line 778: Which “tubulin” are the authors referring to here? There are multiple tubulin genes. This is also an issue in Figure 5’, with “TUB”.’.

Response: A variety of tubulins have been identified to be increased in dystrophic muscles, including tubulin chainsalpha-1B, alpha-1C, alpha-8 and beta-2A. This information has been added to the revised manuscript.

Revised Page 20: ‘… and cadherin-13), (iv) intracellular compensation of dystrophin deficiency by up-regulation of other types of cytoskeletal components (vimentin, desmin, and tubulin chains alpha-1B, alpha-1C, alpha-8 and beta-2A), (v) macrophage invasion and …’.

In revised Figure 5, the term ‘TUB’ has been changed to ‘TUBA/B’.

We would like to thank Reviewer 2 for the constructive criticism of our manuscript and hope that you will find the revised version of our article now acceptable for publication in Proteomes.

Round 2

Reviewer 2 Report

Comments and Suggestions for Authors

Overall, the authors have successfully addressed my concerns and comments.